# Flavors of Margin: Implicit Bias of Steepest Descent in Homogeneous Neural Networks

**Nikolaos Tsilivis**
New York University
nt2231@nyu.edu

**Gal Vardi**
Weizmann Institute of Science
gal.vardi@weizmann.ac.il

**Julia Kempe**
New York University
Meta FAIR

## Abstract

We study the implicit bias of the family of steepest descent algorithms with infinitesimal learning rate, including gradient descent, sign gradient descent and coordinate descent, in deep homogeneous neural networks. We prove that an algorithm-dependent geometric margin increases during training and characterize the late-stage bias of the algorithms. In particular, we define a generalized notion of stationarity for optimization problems and show that the algorithms progressively reduce a (generalized) Bregman divergence, which quantifies proximity to such stationary points of a margin-maximization problem. We then experimentally zoom into the trajectories of neural networks optimized with various steepest descent algorithms, highlighting connections to the implicit bias of Adam.

## 1 Introduction

Overparameterized neural networks excel in many natural supervised learning applications. A theory that aims to explain their strong generalization performance places optimization at the forefront: in problems where many candidate models are available, the optimization algorithm implicitly selects well-generalizing ones (Neyshabur et al., 2015b). The term "implicitly" indicates that neither the unregularized loss function nor the architecture explicitly favors simple, well-generalizing solutions, but this occurs due to the choice of the optimization algorithm. Most existing theoretical results on this so-called *implicit bias of optimization* demonstrate, to some extent, that gradient descent in overparameterized problems biases the solution to be the *simplest*, in terms of the lowest possible $\ell_2$ norm of the weights (Soudry et al., 2018; Ji & Telgarsky, 2019; 2020; Lyu & Li, 2020; Nacson et al., 2019).

Simplicity, however, lies in the eyes of the beholder. For instance, in logistic regression with many irrelevant features, an $\ell_1$-regularized solution is simpler than an $\ell_2$-regularized one (Ng, 2004). Moreover, in contemporary deep learning, Adam (Kingma & Ba, 2015), AdamW (Loshchilov & Hutter, 2019), and related optimization algorithms are preferred for language modeling (Zhang et al., 2020), as their implicit biases seem better suited for such applications than gradient descent. It is therefore important to understand the types of solutions favored by optimization algorithms beyond (stochastic) gradient descent, in order to address the current (and future) applications of deep learning.

In this work, we contribute to this line of research by studying the large family of *steepest descent* algorithms with respect to an arbitrary norm $\|\cdot\|$ in deep, non-linear, homogeneous neural networks. This class of methods generalizes gradient descent to optimization geometries other than the Euclidean, allowing the update rule to operate under a different norm. It includes coordinate descent (which has strong ties to boosting (Mason et al., 1999)) and sign gradient descent (which is closely related to Adam (Kunstner et al., 2023)) as special cases.

**Our contributions.** We provide a unifying, rigorous analysis of any steepest descent algorithm in classification settings with locally-Lipschitz, homogeneous neural networks trained using an exponentially-tailed loss. Specifically, we focus on the late stage of training (after the network

has achieved perfect training accuracy) in the limit of an infinitesimal learning rate. Our first result characterizes the algorithm's tendency to increase an algorithm-dependent margin (Theorem 3.1): similar to prior work on gradient descent (Lyu & Li, 2020), we show that a *soft* version of the geometric margin starts increasing immediately after fitting the training data.

We then turn our attention to the asymptotic properties of the algorithm. In an attempt to find evidence of *margin maximization*, we define a notion of stationary points for optimization problems, which generalizes the usual Karush-Kuhn-Tucker one (Definition 3.4), along with approximate versions (Definition A.9). As we show, during training, the algorithms make implicit progress towards such stationary points of a margin maximization problem in a specific, geometric sense: they progressively reduce a *(generalized) Bregman divergence* (Definition 3.6), which quantifies how well the stationarity condition is satisfied. As this process concludes, the limit points of training are along the direction of a generalized KKT point of the algorithm-dependent geometric margin maximization problem (Theorem 3.8). For algorithms whose squared norm is a smooth function (for example, any $\ell_p$ norm for $p \geq 2$), Theorem 3.8 further implies directional convergence to KKT points of the same margin maximization problem (Corollary 3.8.1).

In total, these results provide evidence for (geometric) margin-maximization in any steepest descent algorithm and significantly generalize prior results that were about gradient descent only (Lyu & Li, 2020; Nacson et al., 2019). Moreover, the generalized divergence can be interpreted as a measure of proximity to stationarity for optimization problems, similarly to what was proposed in prior definitions of approximate KKT points in the literature (Dutta et al., 2013), and could be of broader interest to the optimization community. We find it appealing and theoretically intriguing that, despite the non-convexity of the loss landscape of deep neural networks, such simple convex structures emerge once the data points separate.

Finally, in Section 4, we train neural networks with the three main steepest descent algorithms (gradient descent, sign gradient descent and coordinate descent). We perform experiments in: (a) teacher-student tasks, to assess the connection between implicit bias and generalization and (b) image classification tasks, to study the relationship between Adam and steepest descent algorithms.

## 1.1 RELATED WORK

There have been numerous works studying the implicit biases of optimization in supervised learning and their relationship to geometric margin maximization - see (Vardi, 2023) for a survey.

Steepest descent algorithms with respect to non-Euclidean geometries have been explored before, both in supervised (e.g. (Neyshabur et al., 2015a; Large et al., 2024)) and non-supervised (e.g. (Carlson et al., 2015)) machine learning problems. The implicit bias of this family of optimization methods was first studied in generality in (Gunasekar et al., 2018) in the context of linear models for separable data, where margin maximization was established. Their proof is based on a result on Adaboost due to Telgarsky (2013). Our results generalize the analysis of steepest descent algorithms to any homogeneous neural network. Most related to our paper are the works of Nacson et al. (2019); Lyu & Li (2020) and Ji & Telgarsky (2020). Nacson et al. (2019) studied infinitesimal regularization and its connection to margin maximization in both homogeneous and non-homogeneous deep models, while also proving (directional) convergence of gradient descent to a first order point of an $\ell_2$-margin maximization problem for homogeneous models under strong technical assumptions. Lyu & Li (2020), whose theoretical setup we mainly follow, significantly weakened the assumptions, under which such a result holds, and Lyu & Li (2020) further demonstrated the experimental benefits of margin maximization in terms of robustness. Kunin et al. (2023) generalized these results to a broader class of networks with varying degree of homogeneity, while (Cai et al., 2024) analyzed non-homogeneous 2-layer networks trained with a large learning rate. Vardi et al. (2022) identified cases where the KKT points of the $\ell_2$ margin maximization problems are not (even locally) optimal.

The implicit bias of Adam (Kingma & Ba, 2015) has been previously studied in the works of Wang et al. (2021; 2022) for homogeneous networks, where it is proven that it shares the same asymptotic properties as gradient descent ($\ell_2$ margin maximization). Recently, Zhang et al. (2024) analyzed a version of the algorithm in linear models, without a numerical precision constant, which arguably better captures realistic training runs, and found bias towards $\ell_1$ margin maximization - the same as in the case of sign gradient descent. This makes us optimistic that insights from our analysis, which covers sign gradient descent (steepest descent with respect to the $\ell_\infty$ norm), can shed light on the

poorly understood implicit bias of Adam in deep neural networks. See also (Xie & Li, 2024) for a recently established connection between AdamW (Loshchilov & Hutter, 2019) and sign gradient descent. An additional motivation for studying steepest descent algorithms is in improving the robustness of deep neural networks: Tsilivis et al. (2024), recently, provided experimental evidence and theoretical arguments that deep networks adversarially trained with different steepest descent algorithms exhibit significant differences in their (robust) generalization error.

## 2 BACKGROUND

**Learning Setup** We consider binary classification problems with deep, homogeneous, neural networks. Formally, let $\mathcal{S} = \{\mathbf{x}_i, y_i\}_{i=1}^{m}$ be a dataset of i.i.d. points sampled from an unknown distribution $\mathcal{D}$ with $\mathbf{x}_i \in \mathbb{R}^d$ and $y_i \in \{\pm 1\}$ for all $i \in [m]$, and let $f(\cdot; \boldsymbol{\theta}) : \mathbb{R}^d \to \mathbb{R}$ denote a neural network parameterized by $\boldsymbol{\theta} \in \mathbb{R}^p$. The vector $\boldsymbol{\theta}$ contains all the parameters of the neural network, concatenated into a single vector. We study training under an exponential loss $\mathcal{L}(\boldsymbol{\theta}) = \sum_{i=1}^{m} e^{-y_i f(\mathbf{x}_i; \boldsymbol{\theta})}$. We focus on this setting for simplicity in the main text, but our results should readily generalize to more common losses, such as the logistic loss, as well as its multi-class generalization - the cross-entropy loss. See Section A.3 for details and extensions of our main result.

**Algorithms** The family of steepest descent algorithms generalizes gradient descent to different optimization geometries, allowing the update rule to operate under an arbitrary norm (instead of the usual Euclidean one) (Boyd & Vandenberghe, 2014). Formally, the update rule for *steepest descent* with respect to a norm $\|\cdot\|$ is:

$$\boldsymbol{\theta}_{t+1} = \boldsymbol{\theta}_t + \eta_t \Delta\boldsymbol{\theta}_t, \text{ where } \Delta\boldsymbol{\theta}_t \text{ satisfies}$$
$$\Delta\boldsymbol{\theta}_t = \underset{\|\mathbf{u}\| \leq \|\nabla\mathcal{L}(\boldsymbol{\theta}_t)\|_\star}{\arg\min} \langle \mathbf{u}, \nabla\mathcal{L}(\boldsymbol{\theta}_t) \rangle, \tag{1}$$

where the *dual* norm $\|\cdot\|_\star$ of $\|\cdot\|$ is defined as $\|\mathbf{z}\|_\star = \max_{\mathbf{v}}\{|\langle \mathbf{z}, \mathbf{v} \rangle| : \|\mathbf{v}\| = 1\}$ for any $\mathbf{z}$, and $\eta_t$ is a learning rate. Gradient descent can be derived from Equation 1 with $\|\cdot\| = \|\cdot\|_2$. See Appendix C for details on how steepest descent algorithms are closely related to popular adaptive methods, such as Adam (Kingma & Ba, 2015) and Shampoo (Gupta et al., 2018).

**Assumptions & Technical Points** In order to formally allow commonly used activation functions, such as the ReLU, we theoretically analyze loss landscapes that are not necessarily differentiable. That is, we consider Clarke's subdifferentials (Clarke, 1975) in our analysis:

$$\partial f := \text{conv} \left\{ \lim_{k \to \infty} \nabla f(\mathbf{x}_k) : \mathbf{x}_k \to \mathbf{x}, f \text{ differentiable at } \mathbf{x} \right\}, \tag{2}$$

where $\text{conv}(\cdot)$ stands for the convex hull of a set.
Furthermore, we analyze steepest descent in the limit of infinitesimal step size, i.e. *steepest flow*:

$$\frac{d\boldsymbol{\theta}}{dt} \in \left\{ \underset{\|\mathbf{u}\| \leq \|\mathbf{g}_t\|_\star}{\arg\min} \langle \mathbf{u}, \mathbf{g}_t \rangle : \mathbf{g}_t \in \partial\mathcal{L}(\boldsymbol{\theta}_t) \right\}. \tag{3}$$

This choice simplifies the analysis while still capturing the essence of the biases of the algorithms. Finally, we make the following assumptions:

- (A1) Local Lipschitzness: For any $\mathbf{x}_i \in \mathbb{R}^d$, $f(\mathbf{x}_i; \cdot) : \mathbb{R}^p \to \mathbb{R}$ is locally Lipschitz (and admits a chain rule - see Theorem A.2).
- (A2) $L$-Homogeneity: We assume that $f$ is $L$-homogeneous in the parameters, i.e. $f(\cdot; c\boldsymbol{\theta}) = c^L f(\cdot; \boldsymbol{\theta})$ for any $c > 0$.
- (A3) Realizability. There is a $t_0 > 0$, such that $\mathcal{L}(\boldsymbol{\theta}_{t_0}) < 1$.

Assumption (A1) is a minimal assumption on the regularity of the network, while assumption (A2) includes many commonly used architectures. For instance, ReLU networks with an arbitrary number of layers, but without bias terms, satisfy (A1),(A2). Assumption (A3) ensures that the algorithm will succeed in classifying the training points and allows us to focus on what happens beyond that point of separation. Indeed, we are particularly interested in understanding the geometric properties of the model $f(\cdot; \boldsymbol{\theta}_t)$ as $t \to \infty$ (at convergence) – the so-called *implicit biases* of the learning algorithms.

## 3 THEORY

We analyze the behavior of steepest descent algorithms in the late stage of training and study their geometric properties and how these relate to geometric, algorithm-specific, margins.

### 3.1 ALGORITHM-DEPENDENT MARGIN INCREASES

In *linear* models, where $f(\mathbf{x}; \boldsymbol{\theta}) = \langle \boldsymbol{\theta}, \mathbf{x} \rangle$, the concept of $\|\cdot\|_\star$-*geometric margin* [1], $\min_{i \in [m]} \frac{y_i \langle \boldsymbol{\theta}, \mathbf{x}_i \rangle}{\|\boldsymbol{\theta}\|}$, plays a central and fundamental role in the analysis of the convergence of training (Novikoff, 1963) as well as in the generalization of the final model (Vapnik, 1998). Ideally, we would like to track a similar quantity when training general, homogeneous, non-linear networks $f(\cdot; \boldsymbol{\theta})$ with steepest descent with respect to the $\|\cdot\|$ norm:

$$\gamma(\boldsymbol{\theta}) = \frac{\min_{i \in [m]} y_i f(\mathbf{x}_i; \boldsymbol{\theta})}{\|\boldsymbol{\theta}\|^L} = \min_{i \in [m]} y_i f\left(\mathbf{x}_i; \frac{\boldsymbol{\theta}}{\|\boldsymbol{\theta}\|}\right), \tag{4}$$

where recall that $L$ is the level of homogeneity of the model. As it turns out, it is easier to follow the evolution of the following, *soft*, geometric margin:

$$\widetilde{\gamma}(\boldsymbol{\theta}) = -\frac{\log \mathcal{L}(\boldsymbol{\theta})}{\|\boldsymbol{\theta}\|^L}. \tag{5}$$

The characterisation of "soft" comes from the definition of "softmax" (a.k.a log-sum-exp), which is often used in machine learning. The same idea is used here to approximate the numerator of Equation 4. The soft margin $\widetilde{\gamma}(\boldsymbol{\theta})$ is at most an additive $\log m$ away from $\gamma(\boldsymbol{\theta})$ and converges to $\gamma(\boldsymbol{\theta})$ as $t \to \infty$ - see Lemma A.7 and Corollary A.7.1.

We show next that, given the algorithm has reached a small value in the loss, the soft margin is non-decreasing. This theorem is similar to part of Lemma 5.1 in (Lyu & Li, 2020), which is the key lemma in their result. Ours is admittedly simpler, avoiding a beautiful polar decomposition which was crucial in their analysis, yet, unfortunately, pertinent to the $\ell_2$ case only.

**Theorem 3.1** (Soft margin increases). *For almost any $t > t_0$, it holds:*

$$\frac{d \log \widetilde{\gamma}}{dt} \geq L \left\|\frac{d\boldsymbol{\theta}}{dt}\right\|^2 \left(\frac{1}{L\mathcal{L}(\boldsymbol{\theta}_t) \log \frac{1}{\mathcal{L}(\boldsymbol{\theta}_t)}} - \frac{1}{\|\boldsymbol{\theta}_t\| \left\|\frac{d\boldsymbol{\theta}}{dt}\right\|}\right) \geq 0.$$

*Proof of a simplified version.* We present a proof for a simplified version of this theorem here, covering differentiable networks $f$, while we defer the full proof to Appendix A.2. For differentiable losses, steepest flow corresponds to:

$$\frac{d\boldsymbol{\theta}}{dt} \in \arg\min_{\|\mathbf{u}\| \leq \|\nabla \mathcal{L}(\boldsymbol{\theta}_t)\|_\star} \langle \mathbf{u}, \nabla \mathcal{L}(\boldsymbol{\theta}_t) \rangle. \tag{6}$$

By the definition of the dual norm and chain rule, we have for any $t > 0$:

$$\left\|\frac{d\boldsymbol{\theta}}{dt}\right\| = \|\nabla \mathcal{L}(\boldsymbol{\theta}_t)\|_\star \quad \text{and} \quad \frac{d\mathcal{L}(\boldsymbol{\theta}_t)}{dt} = -\left\|\frac{d\boldsymbol{\theta}}{dt}\right\|^2. \tag{7}$$

---

[1]In this paper, we diverge from the established terminology when it comes to naming margins, by calling it $\|\cdot\|_\star$-geometric margin (instead of $\|\cdot\|$-geometric margin) when it is defined with respect to the $\|\cdot\|$ norm of the parameters. We believe this is proper, since the $\|\cdot\|_\star$-geometric margin in linear models maximizes the metric induced by the $\|\cdot\|_\star$ norm (and not its dual, $\|\cdot\|$).

Let $\mathbf{n}_t \in \partial \|\boldsymbol{\theta}_t\|$ (recall that a norm $\|\cdot\|$ might not be differentiable everywhere). For any $t > t_0$, we have:

$$
\begin{aligned}
\frac{d \log \widetilde{\gamma}}{dt} &= \frac{d}{dt} \log \log \frac{1}{\mathcal{L}(\boldsymbol{\theta}_t)} - L \frac{d}{dt} \log \|\boldsymbol{\theta}_t\| \\
&= \frac{d}{dt} \log \log \frac{1}{\mathcal{L}(\boldsymbol{\theta}_t)} - L \left\langle \frac{\mathbf{n}_t}{\|\boldsymbol{\theta}_t\|}, \frac{d\boldsymbol{\theta}}{dt} \right\rangle \quad \text{(Chain rule)} \\
&\geq \frac{d}{dt} \log \log \frac{1}{\mathcal{L}(\boldsymbol{\theta}_t)} - L \frac{\left\| \frac{d\boldsymbol{\theta}}{dt} \right\|}{\|\boldsymbol{\theta}_t\|} \quad \text{(def. of dual norm and } \|\mathbf{n}_t\|_\star \leq 1, \text{ Lemma A.4)} \\
&= -\frac{d\mathcal{L}(\boldsymbol{\theta}_t)}{dt} \frac{1}{\mathcal{L}(\boldsymbol{\theta}_t) \log \frac{1}{\mathcal{L}(\boldsymbol{\theta}_t)}} - L \frac{\left\| \frac{d\boldsymbol{\theta}}{dt} \right\|}{\|\boldsymbol{\theta}_t\|} \quad \text{(Chain rule)} \\
&= \left\| \frac{d\boldsymbol{\theta}}{dt} \right\|^2 \left( \frac{1}{\mathcal{L}(\boldsymbol{\theta}_t) \log \frac{1}{\mathcal{L}(\boldsymbol{\theta}_t)}} - \frac{L}{\|\boldsymbol{\theta}_t\| \left\| \frac{d\boldsymbol{\theta}}{dt} \right\|} \right). \quad \text{(Equation 7)}
\end{aligned}
\tag{8}
$$

The first term inside the parenthesis can be related to the second one via the following calculation:

$$
\begin{aligned}
\langle \boldsymbol{\theta}_t, -\nabla \mathcal{L}(\boldsymbol{\theta}_t) \rangle &= \left\langle \boldsymbol{\theta}_t, \sum_{i=1}^m e^{-y_i f(\mathbf{x}_i; \boldsymbol{\theta}_t)} y_i \nabla f(\mathbf{x}_i; \boldsymbol{\theta}_t) \right\rangle \\
&= \sum_{i=1}^m e^{-y_i f(\mathbf{x}_i; \boldsymbol{\theta}_t)} y_i \langle \boldsymbol{\theta}_t, \nabla f(\mathbf{x}_i; \boldsymbol{\theta}_t) \rangle \\
&= L \sum_{i=1}^m e^{-y_i f(\mathbf{x}_i; \boldsymbol{\theta}_t)} y_i f(\mathbf{x}_i; \boldsymbol{\theta}_t),
\end{aligned}
\tag{9}
$$

where the last equality follows from Euler's theorem for homogeneous functions. Now, observe that this last term can be lower bounded as:

$$
\langle \boldsymbol{\theta}_t, -\nabla \mathcal{L}(\boldsymbol{\theta}_t) \rangle \geq L \sum_{i=1}^m e^{-y_i f(\mathbf{x}_i; \boldsymbol{\theta}_t)} \min_{i \in [m]} y_i f(\mathbf{x}_i; \boldsymbol{\theta}_t) \geq L \mathcal{L}(\boldsymbol{\theta}_t) \log \frac{1}{\mathcal{L}(\boldsymbol{\theta}_t)},
\tag{10}
$$

where we used the fact $e^{-\min_{i \in [m]} y_i f(\mathbf{x}_i; \boldsymbol{\theta}_t)} \leq \sum_{i=1}^m e^{-y_i f(\mathbf{x}_i; \boldsymbol{\theta}_t)} = \mathcal{L}(\boldsymbol{\theta}_t)$. We have made the first term of Equation 8 appear. By plugging Equation 10 into Equation 8, we get:

$$
\begin{aligned}
\frac{d \log \widetilde{\gamma}}{dt} &\geq \left\| \frac{d\boldsymbol{\theta}}{dt} \right\|^2 \left( \frac{L}{\langle \boldsymbol{\theta}_t, -\nabla \mathcal{L}(\boldsymbol{\theta}_t) \rangle} - \frac{L}{\|\boldsymbol{\theta}_t\| \left\| \frac{d\boldsymbol{\theta}}{dt} \right\|} \right) \\
&\geq \left\| \frac{d\boldsymbol{\theta}}{dt} \right\|^2 \left( \frac{L}{\|\boldsymbol{\theta}_t\| \|\nabla \mathcal{L}(\boldsymbol{\theta}_t)\|_\star} - \frac{L}{\|\boldsymbol{\theta}_t\| \left\| \frac{d\boldsymbol{\theta}}{dt} \right\|} \right). \quad \text{(definition of dual norm)}
\end{aligned}
\tag{11}
$$

Noticing that $\|\nabla \mathcal{L}(\boldsymbol{\theta}_t)\|_\star = \left\| \frac{d\boldsymbol{\theta}}{dt} \right\|$ (from Equation 7) concludes the proof. $\qquad \square$

**Remark 3.2.** *Observe that it is the geometric margin induced by the dual norm of the algorithm that is non-decreasing, and not any geometric margin. The proof crucially relies on the fact that* $\|\nabla \mathcal{L}(\boldsymbol{\theta}_t)\|_\star = \left\| \frac{d\boldsymbol{\theta}}{dt} \right\|$.

### 3.2 CONVERGENCE TO GENERALIZED STATIONARY POINTS OF THE MAX-MARGIN PROBLEM

The previous theorem provides evidence and is a first indication that steepest flow implicitly maximizes the $\|\cdot\|_\star$-geometric margin in deep neural networks. However, the monotonicity of the (soft) margin alone does not imply anything about its final value and its optimality. In this section, we provide a concrete characterization of the *asymptotic* behavior of steepest flow: we show that any limit point of the iterates produced by steepest flow is along the direction of a *generalized KKT* point of the following margin maximization (MM) optimization problem:

$$
\min_{\boldsymbol{\theta} \in \mathbb{R}^d} \frac{1}{2} \|\boldsymbol{\theta}\|^2 \tag{MM}
$$
$$
\text{s.t. } y_i f(\mathbf{x}_i; \boldsymbol{\theta}) \geq 1, \ \forall i \in [m].
$$

Let us recall the definition of a Karush-Kuhn-Tucker point (Karush, 1939; Kuhn, H. W. and Tucker, A. W., 1951).

**Definition 3.3.** *(KKT point) A feasible point $\boldsymbol{\theta} \in \mathbb{R}^p$ of (MM) is a Karush-Kuhn-Tucker (KKT) point, if there exist $\lambda_1, \ldots, \lambda_m \geq 0$ such that:*

1. $\partial \frac{1}{2} \|\boldsymbol{\theta}\|^2 + \sum_{i=1}^m \lambda_i \partial \left(1 - y_i f(\mathbf{x}_i; \boldsymbol{\theta})\right) \ni 0.$

2. $\lambda_i (1 - y_i f(\mathbf{x}_i; \boldsymbol{\theta})) = 0, \ \forall i \in [m].$

Notice that the first so-called *stationarity* condition is defined using set addition, since we are dealing with non-differentiable functions. See (Dutta et al., 2013) for more details on optimization problems with non-smooth objectives/constraints. Under some regularity assumptions, the KKT conditions become necessary conditions for global optimality and for non-convex problems like (MM) they might be the best characterization of optimality we can hope for. See Lemma A.11 for details.

In the following definition of generalized KKT points, we relax the stationarity condition and parameterize it by a non-negative function.

**Definition 3.4.** *(d-generalized KKT point) Let $d : \mathbb{R}^p \times \mathbb{R}^p \to \mathbb{R}_+$. A feasible point $\boldsymbol{\theta} \in \mathbb{R}^p$ of (MM) is called a d-**generalized KKT** point if there exist $\lambda_1, \ldots, \lambda_m \geq 0$, $\mathbf{h}_i \in \partial f(\mathbf{x}_i; \boldsymbol{\theta})$ and $\mathbf{k} \in \partial \frac{1}{2} \|\boldsymbol{\theta}\|^2$ such that:*

1. $d \left( \sum_{i=1}^m \lambda_i y_i \mathbf{h}_i, \mathbf{k} \right) = 0.$

2. $\lambda_i (1 - y_i f(\mathbf{x}_i; \boldsymbol{\theta})) = 0, \ \forall i \in [m].$

**Remark 3.5.** *When $d$ in the definition of a d-generalized KKT point is any metric, we readily recover the original definition of KKT point.*

In Appendix B, we demonstrate how to construct a progress measure for optimization problems leveraging the above notion of stationarity (as well as its approximate version - see Definition A.9).

As we will show, the function $d$, which in our case measures proximity of steepest flow to stationarity, is a generalized *Bregman divergence* induced by the dual norm of the algorithm (squared).

**Definition 3.6.** *(Generalized Bregman divergence) Let $\psi : \mathbb{R}^p \to \mathbb{R}$ with $\psi(\boldsymbol{\theta}) = \frac{1}{2} \|\boldsymbol{\theta}\|_\star^2$ for all $\boldsymbol{\theta} \in \mathbb{R}^p$. We define the (generalized) Bregman divergence $D_{\frac{1}{2}\|\cdot\|_\star^2}^{\mathbf{m}}(\cdot, \cdot) : \mathbb{R}^p \times \mathbb{R}^p \to \mathbb{R}$ induced by $\psi$ as follows:*

$$D_{\frac{1}{2}\|\cdot\|_\star^2}^{\mathbf{m}}(\mathbf{y}, \mathbf{z}) = \frac{1}{2} \|\mathbf{y}\|_\star^2 - \frac{1}{2} \|\mathbf{z}\|_\star^2 - \langle \mathbf{m}, \mathbf{y} - \mathbf{z} \rangle, \tag{12}$$

*where $\mathbf{m} \in \partial \frac{1}{2} \|\mathbf{z}\|_\star^2$.*

**Remark 3.7.** *Notice that if the function $\psi(\boldsymbol{\theta}) = \frac{1}{2} \|\boldsymbol{\theta}\|_\star^2$ is differentiable, then the subdifferential defined at any point collapses to a single element: the gradient of $\psi$. If, further, $\psi$ is strictly convex, then Equation 12 coincides with the usual Bregman divergence induced by $\psi$, defined as $D_\psi(\mathbf{y}, \mathbf{z}) = \psi(\mathbf{y}) - \psi(\mathbf{z}) - \langle \nabla \psi(\mathbf{z}), \mathbf{y} - \mathbf{z} \rangle$. Bregman divergences (Bregman, 1967) generalize the Euclidean squared distance in different geometries and have found numerous applications in machine learning (A. Nemirovskii and D. Yudin, 1983; Banerjee et al., 2005).*

We are, now, ready to state our main result.

**Theorem 3.8.** *Under assumptions (A1), (A2), (A3), consider steepest flow with respect to a norm $\|\cdot\|$ (Equation 3) on the exponential loss $\mathcal{L}(\boldsymbol{\theta}) = \sum_{i=1}^m e^{-y_i f(\mathbf{x}_i; \boldsymbol{\theta})}$. Then, any limit point $\bar{\boldsymbol{\theta}}$ of $\left\{ \frac{\boldsymbol{\theta}_t}{\|\boldsymbol{\theta}_t\|} \right\}_{t \geq 0}$ is along the direction of a $D_{\frac{1}{2}\|\cdot\|_\star^2}^{\widetilde{\boldsymbol{\theta}}}$-generalized KKT point, of the following optimization problem:*

$$\min_{\boldsymbol{\theta}} \frac{1}{2} \|\boldsymbol{\theta}\|^2$$
$$\text{s.t. } y_i f(\mathbf{x}_i; \boldsymbol{\theta}) \geq 1, \ \forall i \in [m], \tag{13}$$

*where $D_{\frac{1}{2}\|\cdot\|_\star^2}^{\widetilde{\boldsymbol{\theta}}}$ is a (generalized) Bregman divergence induced by $\frac{1}{2}\|\cdot\|_\star^2$ and $\widetilde{\boldsymbol{\theta}} = \left( \min_{i \in [m]} y_i f(\mathbf{x}_i; \bar{\boldsymbol{\theta}}) \right)^{-\frac{1}{L}} \bar{\boldsymbol{\theta}}.$*

Theorem 3.8 states that the iterates induced by steepest flow have very specific, geometric properties: not only do they asymptotically approach (in direction) a generalized KKT point of a margin maximization problem, but also, as the proof of this theorem and, in particular, Proposition A.15 tells us, they implicitly make progress towards stationarity by decreasing a Bregman divergence between the gradient of the objective function and the gradient of the constraints of (MM). The full proof can be found in Appendix A. The notion of generalized stationarity (Definition 3.4), as well as its approximate version (Definition A.9) is introduced in order to exactly capture the geometric progress of the algorithm.

While the previous result is not strong enough to guarantee convergence to a KKT point for any case of algorithm norm $\|\cdot\|$, it immediately implies it in the case of a norm whose square is a *smooth* function. We can prove the following corollary for this special class of steepest flows.

**Corollary 3.8.1.** *Under assumptions (A1), (A2), (A3), any limit point $\bar{\theta}$ of $\left\{\frac{\theta_t}{\|\theta_t\|}\right\}_{t\geq 0}$ produced by steepest flow (Equation 3) with respect to a norm $\|\cdot\|$, whose square is a smooth function, on the exponential loss, is along the direction of a KKT point of the optimization problem (MM).*

The proof relies on a fundamental relationship between smoothness of a function and strong convexity of its convex conjugate (Proposition A.19), and can be found in Appendix A. The main contribution of Lyu & Li (2020), which characterizes the implicit bias of gradient flow in homogeneous deep networks, can be recovered by the above result when $\|\cdot\| = \|\cdot\|_2$. Additionally, Corollary 3.8.1 generalizes this result in at least the following cases of algorithm norms:

- Any $\ell_p$ norm with $p \in [2, \infty)$ – see, for example, Lemma 17 in Shalev-Shwartz (2007) for a proof on their smoothness.
- Any norm induced by a positive definite symmetric matrix – i.e. $f(\mathbf{x}) = \langle \mathbf{x}, \mathbf{A}\mathbf{x} \rangle, \mathbf{x} \in \mathbb{R}^d, \mathbf{A} \in \mathbb{R}^{d \times d}$.
- Any $(2, D)$-smooth norm – see Kakade et al. (2008) for details.

To the best of our knowledge, this is a first result about the implicit bias of an algorithm in the parameter space of homogeneous neural networks which is not about $\ell_2$-geometric margin maximization.

## 4 EXPERIMENTS

In this section, we train neural networks with various steepest descent algorithms (gradient descent-`GD`, coordinate descent-`CD`, sign descent-`SD`) to confirm the validity and measure the robustness of the theoretical claims, and to discuss the connection between `Adam` and steepest descent algorithms. Amongst other quantities, we measure the three relevant geometric margins during training, which, in the context of one-hidden layer neural networks with homogeneous activations and without biases, become:

$$\gamma_1 = \min_{i \in [m]} \frac{y_i f(\mathbf{x}_i; \boldsymbol{\theta})}{\|\boldsymbol{\theta}\|_\infty^2}, \quad \gamma_2 = \min_{i \in [m]} \frac{y_i f(\mathbf{x}_i; \boldsymbol{\theta})}{\|\boldsymbol{\theta}\|_2^2}, \quad \gamma_\infty = \min_{i \in [m]} \frac{y_i f(\mathbf{x}_i; \boldsymbol{\theta})}{\|\boldsymbol{\theta}\|_1^2}. \tag{14}$$

### 4.1 STUDENT - TEACHER EXPERIMENTS

We first perform experiments in a controlled environment, where the generative process consists of Gaussian data passed through a one-hidden layer neural network, which is sparse. Specifically:

$$\mathbf{x} \sim \mathcal{N}(0, I_d), \quad y = \text{sgn}\left(f_{\text{teacher}}(\mathbf{x}; \boldsymbol{\theta}^\star)\right) = \text{sgn}\left(\sum_{j=1}^{k} u_j^\star \sigma\left(\langle \mathbf{w}_j^\star, \mathbf{x} \rangle\right)\right), \tag{15}$$

where $\sigma(u) = \max(u, 0)$ is the ReLU activation, $\text{sgn}(\cdot)$ returns the sign of a number, and $\|\boldsymbol{\theta}^\star\|_0$ is assumed to be small. We train neural networks of the same architecture, but of larger width and with randomly initialized weights: $f_{\text{student}}(\mathbf{x}; \boldsymbol{\theta}) = \sum_{j=1}^{k'} u_j \sigma\left(\langle \mathbf{w}_j, \mathbf{x} \rangle\right)$, with width $k' > k$ and $w_{jl} \sim \mathcal{U}\left[-\frac{\alpha}{d}, \frac{\alpha}{d}\right], j \in [k'], l \in [d], u_j \in \mathcal{U}\left[-\frac{\alpha}{k'}, \frac{\alpha}{k'}\right], j \in [k']$ (for `CD` we use: $w_{jl} \sim \mathcal{U}\left[-\frac{\alpha}{k'}, \frac{\alpha}{k'}\right], j \in [k'], l \in [d]$ in order to keep all the individual parameters to the same scale at initialization). The

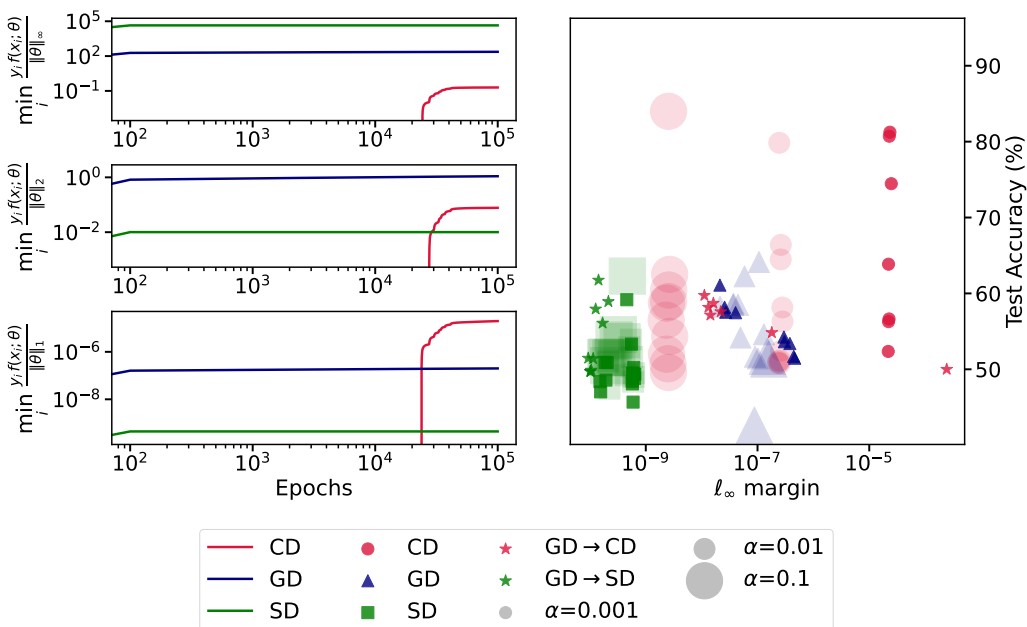

Figure 1: **Evaluation of steepest descent algorithms in a teacher-student setup.** *Left:* Geometric margins ($\gamma_1, \gamma_2, \gamma_\infty$ in Equation 14) over the course of training (average over 20 different seeds). *Right:* Final test accuracy vs final $\ell_\infty$ margin ($\gamma_\infty$). Each point in the 2d space corresponds to a different run (only showing runs that did not diverge). Larger points correspond to larger initialization scales $\alpha$. The star points are produced by switching from GD to CD (red) or SD (green), right after the point of perfect train accuracy.

magnitude of initialization $\alpha$ can control how fast the implicit bias of the algorithm kicks in, with smaller values entering this so-called "rich" regime faster (Woodworth et al., 2020). We compare the performance of (full batch) GD, CD and SD in minimizing the empirical, exponential, loss, consisting of $m$ independent points sampled from the generative process of Equation 15. Section D contains full experimental details.

According to Theorems 3.1, 3.8, we expect GD to favor solutions with small $\ell_2$ norm. This is equivalent to a small sum of the product of the magnitude of incoming and outcoming weights across all neurons (Theorem 1 in (Neyshabur et al., 2015b)). On the other hand, CD will seek to minimize the $\ell_1$ norm of the parameters, which translates to a narrow network with sparse 1st-layer weights. Finally, SD's bias towards small $\|\boldsymbol{\theta}\|_\infty$ solutions does not appear to be useful for generalizing from few samples in this task. Therefore, we expect CD > GD > SD in terms of generalization.

Figure 1 displays our main findings. We summarize our key findings below:

(i) **Margins increase past $t_0$.** As expected from Lemma 3.1, we observe that, right after the point of separation, each algorithm implicitly increases its corresponding geometric margin (Figure 1 left). Furthermore, we observe that the ordering of the algorithms is as expected for each margin (SD attains larger $\ell_1$ margin than GD and CD, etc.), despite the fact that Theorem 3.8 only guarantees convergence to a KKT point (at best) of the margin maximization problem - note the log-log plot.

(ii) **Smaller initialization produces larger geometric margin.** A smaller magnitude of initialization $\alpha$ causes a larger eventual value of the geometric margin (see Figure 1 right for CD and $\gamma_\infty$, where this effect is stronger, and Figure 4 in Appendix D for $\gamma_1, \gamma_2$).

(iii) **Importance of early-stage dynamics for generalization.** Figure 1 right shows the final test accuracy of the networks (20 different runs) vs the value of their final $\ell_\infty$ margin ($\gamma_\infty$). We observe that, while there exist more CD runs with good generalization (red circles), these do not always coincide with larger $\gamma_\infty$. Furthermore, intervening in the algorithms to encourage or discourage $\gamma_\infty$-maximization does not result in significant generalization

changes: after running `GD` until the point of perfect train accuracy, we switch to either `SD` (green stars) or `CD` (red stars) to directly control the late stage geometric properties of the model. Switching to `CD` seems to bear marginal benefits in terms of generalization, even though all the switched runs reach smaller values of $\ell_\infty$ margins compared to the full, no-switching, `GD` runs. These benefits, however, pale in comparison to the full `CD` runs. Switching to `SD`, on the other hand, results in smaller $\gamma_\infty$ and similar or marginally worse test accuracy. See also Figure 4 for test accuracy vs the other two geometric margins. We conclude that it is unlikely that large generalization benefits can solely and causally be linked to larger geometric margins in this setup, and it appears that the early stage dynamics play an important role for generalization.

## 4.2 CONNECTION BETWEEN ADAM AND SIGN-GD

Adaptive optimization methods like `Adam` (Kingma & Ba, 2015) have been popular in deep learning applications, yet theoretically their value has been questioned (Wilson et al., 2017) and their properties remain poorly understood. Wang et al. (2021; 2022) studied the implicit bias of `Adam` in homogeneous networks and concluded that `Adam` shares the same asymptotic properties as `GD`. More recently, this conclusion has been challenged, in the sense that this asymptotic property crucially depends on a precision parameter of the algorithm and does not capture realistic runs of the algorithm (see Section C.1 for details). In particular, it was shown that, in linear models, `Adam`, without this precision parameter, implicitly maximizes the $\ell_1$-geometric margin (Zhang et al., 2024), a property shared with `SD` and not `GD`. Indeed, `Adam` without momentum, and ignoring the precision parameters, is equivalent to `SD`. Setting the precision parameter to 0, on the other hand, is not useful in practical applications, as small initial values of the gradient result in divergence of the loss. A question arises: what, then, are the relevant geometric properties of `Adam` *in practice*?

Figure 2 provides some experimental answers to this question, in light of Theorems 3.1, 3.8. We train two-layer neural networks on a pair of digits extracted from MNIST with `GD`, `SD` and `Adam`, with small initialization. See Section D for experimental details. We observe that, as soon as the algorithms reach 100% train accuracy, the margins start to increase (as Theorem 3.1 suggests); `SD` reaches a larger value of $\gamma_1$, while `GD` reaches a larger value of $\gamma_2$. Interestingly, `Adam` with the default hyperparameters (precision $\epsilon = 10^{-8}$ and non-zero momentum), initially, behaves similar to `SD`, increasing $\gamma_1$, before it starts decreasing it, in order to slowly start increasing $\gamma_2$! Curiously, larger values of $\epsilon$ increase $\gamma_1$ even further and start the second phase slower, but more aggressively. Notice, however, that train and test accuracies have long converged, so it is unlikely that a typical run would have lasted long enough to see the second phase of $\ell_2$-margin maximization (in particular, the loss value needs to be smaller than $10^{-7}$ in order to observe such behavior). Similar observations hold for `Adam` without momentum (recall that without momentum and for $\epsilon \to 0$, we recover `SD`). Therefore, it appears that the $\ell_1$ bias of `SD` (Theorems 3.1, 3.8 for $\|\cdot\| = \|\cdot\|_\infty$) more faithfully describes a typical run of `Adam` in neural networks.

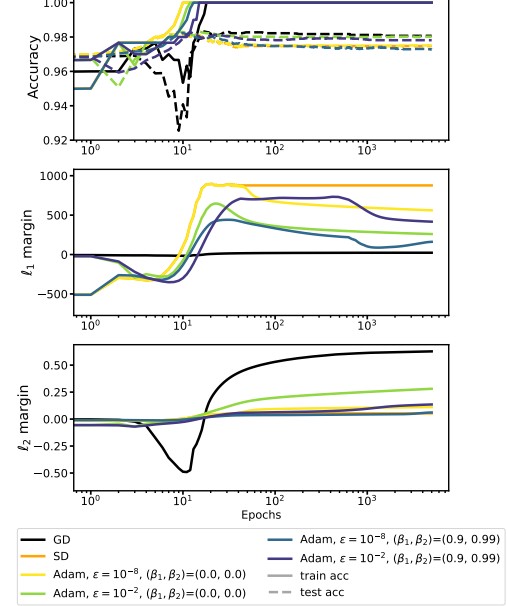

Figure 2: **Relationship between Adam and steepest descent algorithms.** Accuracy, and $\ell_1, \ell_2$ margins during training for `GD`, `SD` and `Adam` on MNIST (3 random seeds). `Adam` is parameterized by a numerical precision constant $\epsilon$ and two momentum parameters $(\beta_1, \beta_2)$ (defaulting to $10^{-8}$ and $(0.9, 0.99)$, respectively). We observe that `Adam` behaves similar to `SD` for the period right after the point of perfect train accuracy.

## 5 CONCLUSION

In our work, we considered the large family of steepest descent algorithms with respect to an arbitrary norm $\|\cdot\|$ and provided a unifying theoretical analysis of their late-stage implicit bias when training homogeneous neural networks. Furthermore, we introduced a notion of stationarity for optimization problems (defined with respect to a Bregman divergence induced by the algorithm norm), which, as we showed, captures the implicit progress of the algorithms and might be of independent interest. Theorem 3.8 does not preclude the possibility that any steepest descent algorithm will converge to a KKT point; yet our positive result (Corollary 3.8.1) shows this in the case where the algorithm squared norm is smooth. It would be interesting to generalize this result to any norm or show a counterexample, as well as to generalize our proof for a discrete time analysis.

Our results can reinforce several recent efforts that attempt to understand deep learning through the lens of implicit bias. In particular, questions about generalization, robustness, and privacy can now be asked more broadly: (a) can we extract training samples from neural networks optimized with Adam, leveraging its connection to sign gradient descent, in a similar fashion to what has been shown to be possible for gradient descent (Haim et al., 2022)? (b) can we leverage our implicit bias results to design more sample-efficient algorithms for robust training, as argued by Tsilivis et al. (2024)? (c) is benign overfitting a general property of first-order methods, or are current results (e.g. (Frei et al., 2022; Shamir, 2023)) specifically tailored for gradient descent?

ACKNOWLEDGEMENTS

NT and JK acknowledge support through the NSF NRT training grant award 1922658. GV is supported by a research grant from Mortimer Zuckerman, the Zuckerman STEM Leadership Program, and by research grants from the Center for New Scientists at the Weizmann Institute of Science, and the Shimon and Golde Picker – Weizmann Annual Grant. We would like to thank Zhiyuan Li and Kaifeng Lyu for a useful discussion. Part of this work was done while NT was visiting the Toyota Technological Institute of Chicago (TTIC) during the winter of 2024, and NT would like to thank everyone at TTIC for their hospitality, which enabled this work. Part of this work was done while JK and NT were hosted by the Centre Sciences de Donnees at the École Normale Supérieure (ENS) in 2023/24, whose hospitality we gratefully acknowledge. This work was supported in part through the NYU IT High Performance Computing resources, services, and staff expertise.

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

## A  MISSING PROOFS

In this section, we provide proofs for the results stated in the main text.

### A.1  STEEPEST FLOW

We first present a series of technical results, which are about steepest flow in the case of non-differentiable loss functions. In what follows, we will denote with $\mathbf{g}_t^\star$ any loss subderivative with minimum $\|\cdot\|_\star$ norm, i.e. $\mathbf{g}_t^\star \in \arg\min_{\mathbf{u} \in \partial\mathcal{L}(\boldsymbol{\theta}_t)} \|\mathbf{u}\|_\star$. In the case of subdifferentials, chain rule holds as an inclusion:

**Theorem A.1** (Theorem 2.3.9 and 2.3.10 in Clarke (1990)). *Let $z_1, \ldots, z_n : \mathbb{R}^d \to \mathbb{R}$, $f : \mathbb{R}^n \to \mathbb{R}$ be locally Lipschitz functions and define $\mathbf{z} = (z_1, \ldots, z_n)$. Let $(f \circ \mathbf{z})(\mathbf{x}) = f(z_1(\mathbf{x}), \ldots, z_n(\mathbf{x}))$ be the composition of $\mathbf{z}$ with $f$. Then, it holds:*

$$\partial(f \circ \mathbf{z})(\mathbf{x}) \subseteq \text{conv} \left\{ \sum_{i=1}^n \alpha_i \mathbf{h}_i : \boldsymbol{\alpha} \in \partial f(z_1(\mathbf{x}), \ldots, z_n(\mathbf{x})), \mathbf{h}_i \in \partial z_i(\mathbf{x}) \right\}. \tag{16}$$

To further analyze steepest flows and to guarantee loss monotonicity, we need a stronger chain rule result. This can be achieved for a large class of locally Lipschitz functions, as per the following theorem which is due to Davis et al. (2020).

**Theorem A.2.** *(Theorem 5.8 in Davis et al. (2020)) If $F : \mathbb{R}^k \to \mathbb{R}$ is locally Lipschitz and Whitney $C^1$-stratifiable, then it admits a chain rule: for all arcs (functions which are absolutely continuous on every compact subinterval) $\mathbf{u} : [0, \infty) \to \mathbb{R}^k$, almost all $t \geq 0$, and all $\mathbf{g} \in \partial F(\mathbf{u}(t))$, it holds:*

$$\frac{dF(\mathbf{u}(t))}{dt} = \left\langle \mathbf{g}, \frac{d\mathbf{u}(t)}{dt} \right\rangle. \tag{17}$$

Whitney $C^1$-stratifiability includes a large family of functions, including functions defined in an o-minimal structure which has been a standard assumption in the literature Ji & Telgarsky (2019). It excludes some pathological functions - see, for instance, Appendix J in Lyu & Li (2020). This version of chain rule allows us to derive the following central properties of steepest flows.

**Proposition A.3.** *Let $\mathcal{L} : \mathbb{R}^p \to \mathbb{R}$ and assume that $\mathcal{L}$ admits a chain rule. Then, for the steepest flow iterates of Equation 1, it holds for almost any $t \geq 0$:*

$$\frac{d\mathcal{L}}{dt} = -\left\| \frac{d\boldsymbol{\theta}}{dt} \right\|^2 \leq 0, \tag{18}$$

*and*

$$\left\langle \frac{d\boldsymbol{\theta}}{dt}, -\mathbf{g}_t^\star \right\rangle = \left\| \frac{d\boldsymbol{\theta}}{dt} \right\|^2 = \|\mathbf{g}_t^\star\|_\star^2, \tag{19}$$

*where $\mathbf{g}_t^\star \in \arg\min_{\mathbf{u} \in \partial\mathcal{L}(\boldsymbol{\theta}_t)} \|\mathbf{u}\|_\star$.*

*Proof.* From Theorem A.2, for almost any $t \geq 0$, it holds $\forall \, \mathbf{g}_t \in \partial\mathcal{L}(\boldsymbol{\theta}_t)$:

$$\frac{d\mathcal{L}}{dt} = \left\langle \mathbf{g}_t, \frac{d\boldsymbol{\theta}}{dt} \right\rangle. \tag{20}$$

Applying this for the element of $\partial\mathcal{L}(\boldsymbol{\theta}_t)$, $\mathbf{g}_t'$, that corresponds to $\frac{d\boldsymbol{\theta}}{dt}$ from the definition of steepest flow Equation 3, we get:

$$\frac{d\mathcal{L}}{dt} = \left\langle \mathbf{g}_t', \frac{d\boldsymbol{\theta}}{dt} \right\rangle = -\left\| \frac{d\boldsymbol{\theta}}{dt} \right\|^2, \tag{21}$$

where the last equality follows from the definition of the dual norm. But, Equation 20 for $\mathbf{g}_t^\star \in \arg\min_{\mathbf{u} \in \partial\mathcal{L}(\boldsymbol{\theta}_t)} \|\mathbf{u}\|_\star$, yields:

$$\left| \frac{d\mathcal{L}}{dt} \right| = \left| \left\langle \mathbf{g}_t^\star, \frac{d\boldsymbol{\theta}}{dt} \right\rangle \right| \leq \|\mathbf{g}_t^\star\|_\star \left\| \frac{d\boldsymbol{\theta}}{dt} \right\|. \tag{22}$$

Thus, combining Equation 21, Equation 22, we obtain:

$$\left\| \frac{d\boldsymbol{\theta}}{dt} \right\| \leq \|\mathbf{g}_t^\star\|_\star, \tag{23}$$

which implies that the update rule Equation 3 iq equivalent to:

$$\frac{d\boldsymbol{\theta}}{dt} \in \left\{ \underset{\|\mathbf{u}\| \leq \|\mathbf{g}_t\|_\star}{\arg\min} \langle \mathbf{u}, \mathbf{g}_t \rangle : \mathbf{g}_t^\star \in \underset{\mathbf{u} \in \partial \mathcal{L}(\boldsymbol{\theta}_t)}{\arg\min} \|\mathbf{u}\|_\star \right\}. \tag{24}$$

Therefore, from the definition of the dual norm, we have:

$$\left\langle \frac{d\boldsymbol{\theta}}{dt}, -\mathbf{g}_t^\star \right\rangle = \left\| \frac{d\boldsymbol{\theta}}{dt} \right\|^2 = \|\mathbf{g}_t^\star\|_\star^2. \tag{25}$$

$$\square$$

Hence, under the mild assumptions of Theorem A.2, the loss is non-increasing during training.

## A.2 LATE PHASE IMPLICIT BIAS

A useful standard characterization of the subdifferential of a norm is the following:

**Lemma A.4.**
$$\partial\|\mathbf{x}\| = \{\mathbf{v} : \langle \mathbf{v}, \mathbf{x} \rangle = \|\mathbf{x}\|, \|\mathbf{v}\|_\star \leq 1\}$$

We present the proofs for our results about the late stage of training in steepest flow algorithms. The next lemma quantifies the behavior of the smooth margin past the point $t_0$ (where, recall, zero classification error is achieved).

**Theorem A.5** (Soft margin increases - full version). *For almost any $t > t_0$, it holds:*

$$\frac{d\log\widetilde{\gamma}}{dt} \geq L \left\| \frac{d\boldsymbol{\theta}}{dt} \right\|^2 \left( \frac{1}{L\mathcal{L}(\boldsymbol{\theta}_t)\log\frac{1}{\mathcal{L}(\boldsymbol{\theta}_t)}} - \frac{1}{\|\boldsymbol{\theta}_t\| \left\|\frac{d\boldsymbol{\theta}}{dt}\right\|} \right) \geq 0.$$

*Proof.* Let $\mathbf{n}_t \in \partial\|\boldsymbol{\theta}_t\|$. We have:

$$
\begin{aligned}
\frac{d\log\widetilde{\gamma}}{dt} &= \frac{d}{dt}\log\log\frac{1}{\mathcal{L}(\boldsymbol{\theta}_t)} - L\frac{d}{dt}\log\|\boldsymbol{\theta}_t\| \\
&= \frac{d}{dt}\log\log\frac{1}{\mathcal{L}(\boldsymbol{\theta}_t)} - L\left\langle \frac{\mathbf{n}_t}{\|\boldsymbol{\theta}_t\|}, \frac{d\boldsymbol{\theta}}{dt} \right\rangle \quad \text{(Chain rule)} \\
&\geq \frac{d}{dt}\log\log\frac{1}{\mathcal{L}(\boldsymbol{\theta}_t)} - L\frac{\left\|\frac{d\boldsymbol{\theta}}{dt}\right\|}{\|\boldsymbol{\theta}_t\|} \quad \text{(definition of dual norm and } \|\mathbf{n}_t\|_\star \leq 1\text{)} \\
&= -\frac{d\mathcal{L}(\boldsymbol{\theta}_t)}{dt}\frac{1}{\mathcal{L}(\boldsymbol{\theta}_t)\log\frac{1}{\mathcal{L}(\boldsymbol{\theta}_t)}} - L\frac{\left\|\frac{d\boldsymbol{\theta}}{dt}\right\|}{\|\boldsymbol{\theta}_t\|} \quad \text{(Chain rule)} \\
&= \left\| \frac{d\boldsymbol{\theta}}{dt} \right\|^2 \left( \frac{1}{\mathcal{L}(\boldsymbol{\theta}_t)\log\frac{1}{\mathcal{L}(\boldsymbol{\theta}_t)}} - \frac{L}{\|\boldsymbol{\theta}_t\| \left\|\frac{d\boldsymbol{\theta}}{dt}\right\|} \right) \quad \text{(eq. Equation 18).}
\end{aligned}
\tag{26}
$$

But, the first term inside the parenthesis can be related to the second one via the following calculation. Recall that, by Theorem A.2, for any $\mathbf{g}_t \in \partial\mathcal{L}(\boldsymbol{\theta}_t)$ there exist $\mathbf{h}_1 \in \partial y_1 f(\mathbf{x}_1; \boldsymbol{\theta}_t), \ldots, \mathbf{h}_m \in \partial y_m f(\mathbf{x}_m; \boldsymbol{\theta}_t)$ such that $\mathbf{g}_t = \sum_{i=1}^m e^{-y_i f(\mathbf{x}_i; \boldsymbol{\theta}_t)}\mathbf{h}_i$. Thus, for a minimum norm subderivative $\mathbf{g}_t^\star$, we have:

$$
\begin{aligned}
\langle \boldsymbol{\theta}_t, -\mathbf{g}_t^\star \rangle &= \left\langle \boldsymbol{\theta}_t, \sum_{i=1}^m e^{-y_i f(\mathbf{x}_i; \boldsymbol{\theta}_t)}\mathbf{h}_i^\star \right\rangle \\
&= \sum_{i=1}^m e^{-y_i f(\mathbf{x}_i; \boldsymbol{\theta}_t)} \langle \boldsymbol{\theta}_t, \mathbf{h}_i^\star \rangle \\
&= L\sum_{i=1}^m e^{-y_i f(\mathbf{x}_i; \boldsymbol{\theta}_t)} y_i f(\mathbf{x}_i; \boldsymbol{\theta}_t),
\end{aligned}
\tag{27}
$$

where the last equality follows from Euler's theorem for homogeneous functions (whose generalization for subderivatives can be found in Theorem B.2 in Lyu & Li (2020)). Now, observe that this last term can be lower bounded as:

$$\langle \boldsymbol{\theta}_t, -\mathbf{g}_t^\star \rangle \geq L \sum_{i=1}^m e^{-y_i f(\mathbf{x}_i; \boldsymbol{\theta}_t)} \min_{i \in [m]} y_i f(\mathbf{x}_i; \boldsymbol{\theta}_t) \geq L\mathcal{L}(\boldsymbol{\theta}_t) \log \frac{1}{\mathcal{L}(\boldsymbol{\theta}_t)}, \tag{28}$$

where we used the fact $e^{-\min_{i \in [m]} y_i f(\mathbf{x}_i; \boldsymbol{\theta}_t)} \leq \sum_{i=1}^m e^{-y_i f(\mathbf{x}_i; \boldsymbol{\theta}_t)}$. We have made the first term of eq. Equation 68 appear. By plugging eq. Equation 28 into eq. Equation 68, we get:

$$
\begin{aligned}
\frac{d \log \widetilde{\gamma}}{dt} &\geq \left\| \frac{d\boldsymbol{\theta}}{dt} \right\|^2 \left( \frac{L}{\langle \boldsymbol{\theta}_t, -\mathbf{g}_t^\star \rangle} - \frac{L}{\|\boldsymbol{\theta}_t\| \left\| \frac{d\boldsymbol{\theta}}{dt} \right\|} \right) \\
&\geq \left\| \frac{d\boldsymbol{\theta}}{dt} \right\|^2 \left( \frac{L}{\|\boldsymbol{\theta}_t\| \|\mathbf{g}_t^\star\|_\star} - \frac{L}{\|\boldsymbol{\theta}_t\| \left\| \frac{d\boldsymbol{\theta}}{dt} \right\|} \right) \quad \text{(definition of dual norm).}
\end{aligned} \tag{29}
$$

Noticing that $\|\mathbf{g}_t^\star\|_\star = \left\| \frac{d\boldsymbol{\theta}}{dt} \right\|$ (from eq. Equation 19) concludes the proof. $\square$

By extending Lemma B.6 of Lyu & Li (2020), we can further prove that the loss converges to 0 and, thus, the norm of the iterates diverges to infinity.

**Lemma A.6.** *As $t \to \infty$, $\mathcal{L}(\boldsymbol{\theta}_t) \to 0$ and $\|\boldsymbol{\theta}_t\| \to \infty$.*

*Proof.* We suppress the dependence of the loss and the iterates from time $t$, when it is obvious from the context.

From the definition of the steepest flow update and chain rule (eq. Equation 18), we have

$$-\frac{d\mathcal{L}}{dt} = \left\| \frac{d\boldsymbol{\theta}}{dt} \right\|^2 = \|\mathbf{g}_t^\star\|_\star^2 \geq \frac{1}{\|\boldsymbol{\theta}\|^2} \langle \boldsymbol{\theta}, -\mathbf{g}_t^\star \rangle^2, \tag{30}$$

where we applied eqs. Equation 19, Equation 18 and the definition of the dual norm. But, as we showed in eq. Equation 28, the above inner product can be upper bounded by a function of the loss, so, by plugging in, we get:

$$-\frac{d\mathcal{L}}{dt} \geq \frac{L^2}{\|\boldsymbol{\theta}\|^2} \left( \mathcal{L} \log \frac{1}{\mathcal{L}} \right)^2 = \frac{L^2}{\left(\log \frac{1}{\mathcal{L}}\right)^{2/L}} \widetilde{\gamma}^{2/L}(t) \left( \mathcal{L} \log \frac{1}{\mathcal{L}} \right)^2 \geq \frac{L^2}{\left(\log \frac{1}{\mathcal{L}}\right)^{2/L}} \widetilde{\gamma}^{2/L}(t_0) \left( \mathcal{L} \log \frac{1}{\mathcal{L}} \right)^2, \tag{31}$$

which follows from the definition of the margin Equation 5 and its monotonicity (Lemma A.5). By rearranging:

$$-\frac{d\mathcal{L}}{dt} \frac{1}{\mathcal{L}^2} \left( \log \frac{1}{\mathcal{L}} \right)^{2/L-2} \geq L^2 \widetilde{\gamma}(t_0)^{2/L}, \tag{32}$$

and integrating over time from $t_0$ to $t > t_0$, we further have:

$$\int_{t_0}^t \left( \log \frac{1}{\mathcal{L}} \right)^{2/L-2} \frac{d}{dt} \frac{1}{\mathcal{L}} dt \geq L^2 \widetilde{\gamma}(t_0)^{2/L} (t - t_0), \tag{33}$$

or, by a change of variables,

$$\int_{1/\mathcal{L}(t_0)}^{1/\mathcal{L}(t)} (\log u)^{2/L-2} \, du \geq L^2 \widetilde{\gamma}(t_0)^{2/L} (t - t_0). \tag{34}$$

The RHS diverges to infinity as $t \to \infty$, hence so does the LHS, which can only happen if $\mathcal{L} \to 0$. In order for $\mathcal{L}(\boldsymbol{\theta}_t) = \sum_{i=1}^m e^{-y_i f(\mathbf{x}_i; \boldsymbol{\theta}_t)} = \sum_{i=1}^m e^{-y_i \|\boldsymbol{\theta}_t\|^L f\left(\mathbf{x}_i; \frac{\boldsymbol{\theta}_t}{\|\boldsymbol{\theta}_t\|}\right)}$ to go to zero, it must be $\|\boldsymbol{\theta}_t\| \to \infty$. $\square$

The following Lemma quantifies the connection between soft and hard margin.

**Lemma A.7.** *For any $\boldsymbol{\theta}$, it holds:*

$$\frac{\min_{i \in [m]} y_i f(\mathbf{x}_i; \boldsymbol{\theta}) - \log m}{\|\boldsymbol{\theta}\|^L} \leq \widetilde{\gamma} \leq \frac{\min_{i \in [m]} y_i f(\mathbf{x}_i; \boldsymbol{\theta})}{\|\boldsymbol{\theta}\|^L}. \tag{35}$$

*Proof.* Follows from:

$$e^{-\min_{i \in [m]} y_i f(\mathbf{x}_i; \boldsymbol{\theta})} \le \mathcal{L}(\boldsymbol{\theta}) \le m e^{-\min_{i \in [m]} y_i f(\mathbf{x}_i; \boldsymbol{\theta})}. \tag{36}$$

$\square$

From the previous two Lemmata, we deduce that the soft margin converges to the hard margin as $t \to \infty$.

**Corollary A.7.1.** *For any $t > t_0$, $\boldsymbol{\theta}_t \in \mathbb{R}^p$, let $\gamma(\boldsymbol{\theta}_t) = \frac{\min_{i \in [m]} y_i f(\mathbf{x}_i; \boldsymbol{\theta}_t)}{\|\boldsymbol{\theta}_t\|^L}$. Then, it holds:*

$$\lim_{t \to \infty} |\widetilde{\gamma}(\boldsymbol{\theta}_t) - \gamma(\boldsymbol{\theta}_t)| = 0. \tag{37}$$

*Proof.* By taking limits in Equation 35, we have:

$$\lim_{t \to \infty} \frac{\min_{i \in [m]} y_i f(\mathbf{x}_i; \boldsymbol{\theta})}{\|\boldsymbol{\theta}\|^L} - \frac{\log m}{\|\boldsymbol{\theta}\|^L} \le \lim_{t \to \infty} \widetilde{\gamma}(\boldsymbol{\theta}_t) \le \lim_{t \to \infty} \frac{\min_{i \in [m]} y_i f(\mathbf{x}_i; \boldsymbol{\theta})}{\|\boldsymbol{\theta}\|^L} \iff$$
$$\lim_{t \to \infty} \gamma(\boldsymbol{\theta}_t) - \lim_{t \to \infty} \frac{\log m}{\|\boldsymbol{\theta}\|^L} \le \lim_{t \to \infty} \widetilde{\gamma}(\boldsymbol{\theta}_t) \le \lim_{t \to \infty} \gamma(\boldsymbol{\theta}_t). \tag{38}$$

But, from Lemma A.6, we know that $\|\boldsymbol{\theta}_t\| \to \infty$. Thus,

$$\lim_{t \to \infty} \gamma(\boldsymbol{\theta}_t) \le \lim_{t \to \infty} \widetilde{\gamma}(\boldsymbol{\theta}_t) \le \lim_{t \to \infty} \gamma(\boldsymbol{\theta}_t), \tag{39}$$

which proves the claim. $\square$

The last part of the proof consists of characterizing the (directional) convergence of the iterates in relation to stationary points of the following optimization problem (re-introduced here for convenience):

$$\min_{\boldsymbol{\theta}} \frac{1}{2} \|\boldsymbol{\theta}\|^2$$
$$\text{s.t. } y_i f(\mathbf{x}_i; \boldsymbol{\theta}) \ge 1, \ \forall i \in [m]. \tag{40}$$

Under some regularity assumptions, the KKT conditions (Definition 3.3) become necessary for global optimality (yet, not sufficient):

**Definition A.8.** *We say that a feasible point of Equation 40 satisfies the Mangasarian-Fromovitz Constraint Qualifications if there exists $\mathbf{v} \in \mathbb{R}^p$ such that for all $i \in [m]$ with $1 - y_i f(\mathbf{x}_i; \boldsymbol{\theta}) = 0$ and for all $\mathbf{h} \in \partial (1 - y_i f(\mathbf{x}_i; \boldsymbol{\theta}))$, it holds:*

$$\langle \mathbf{v}, \mathbf{h} \rangle > 0. \tag{41}$$

Our proof uses the following relaxed notion of stationarity.

**Definition A.9.** *($(d, \epsilon, \delta)$- approximate KKT point) Let $d : \mathbb{R}^p \times \mathbb{R}^p \to \mathbb{R}_+$. A feasible point $\boldsymbol{\theta}$ of equation 40 is called an $(d, \epsilon, \delta)-$approximate KKT point if there exist $\lambda_1, \dots, \lambda_m \ge 0$, $\mathbf{h}_i \in \partial f(\mathbf{x}_i; \boldsymbol{\theta})$ and $\mathbf{k} \in \partial \frac{1}{2} \|\boldsymbol{\theta}\|^2$ such that:*

*1. $d \left( \sum_{i=1}^m \lambda_i y_i \mathbf{h}_i, \mathbf{k} \right) \le \epsilon$*

*2. $\sum_{i=1}^m \lambda_i (y_i f(\mathbf{x}_i; \boldsymbol{\theta}) - 1) \le \delta$.*

We first show that we can always construct a feasible point of Equation 40 from a scaled version of $\boldsymbol{\theta}_t$.

**Lemma A.10.** *For any $t > 0$, $\widetilde{\boldsymbol{\theta}}_t = \frac{\boldsymbol{\theta}_t}{\left( \min_{i \in [m]} y_i f(\mathbf{x}_i; \boldsymbol{\theta}_t) \right)^{\frac{1}{L}}}$ is a feasible point of Equation 40.*

*Proof.* From the homogeneity of $f$, we have:

$$y_i f(\mathbf{x}_i; \widetilde{\boldsymbol{\theta}}_t) = y_i f \left( \mathbf{x}_i; \frac{\boldsymbol{\theta}_t}{\left( \min_{i \in [m]} y_i f(\mathbf{x}_i; \boldsymbol{\theta}_t) \right)^{\frac{1}{L}}} \right) = \frac{y_i f(\mathbf{x}_i; \boldsymbol{\theta}_t)}{\min_{i \in [m]} y_i f(\mathbf{x}_i; \boldsymbol{\theta}_t)} \ge 1 \tag{42}$$

for all $i \in [m]$. So $\widetilde{\boldsymbol{\theta}}_t$ is a feasible point of Equation 40. $\square$

The next Lemma shows that Problem 40 satisfies the Mangasarian-Fromovitz Constraint Qualifications:

**Lemma A.11.** *Problem 40 satisfies the Mangasarian-Fromovitz Constraint Qualifications at every feasible point $\boldsymbol{\theta}$.*

*Proof.* Let $\mathbf{h}_i \in \partial(1 - y_i f(\mathbf{x}_i; \boldsymbol{\theta}))$ and $\mathbf{v} = -\boldsymbol{\theta}$, then for all $i \in [m]$ satisfying $y_i f(\mathbf{x}_i; \boldsymbol{\theta}) = 1$, we have from Euler's theorem for homogeneous functions:

$$\langle \mathbf{v}, \mathbf{h}_i \rangle = L y_i f(\mathbf{x}_i; \boldsymbol{\theta}) = L > 0. \tag{43}$$

$\square$

Our proof uses core ideas from the theory of conjugate functions and Fenchel's duality.

**Definition A.12** (Convex conjugate). *Let $\psi : \mathbb{R}^p \to \mathbb{R}$. We denote by $\psi^\star(\cdot)$ the convex conjugate of $\psi(\cdot)$:*

$$\psi^\star(\boldsymbol{\omega}) = \sup_{\boldsymbol{\theta} \in \mathbb{R}^p} \{\langle \boldsymbol{\omega}, \boldsymbol{\theta} \rangle - \psi(\boldsymbol{\theta})\}. \tag{44}$$

We will make use of the following properties of conjugate functions.

**Proposition A.13.** *(Conjugate subgradient theorem - Theorem 23.5 in Rockafellar (1970), Theorem 4.20 in Beck (2017)) Let $\psi : \mathbb{R}^p \to \mathbb{R}$ be convex and closed. For any $\boldsymbol{\theta}^\star \in \partial \psi^\star(\boldsymbol{\theta})$, it holds $\partial \psi(\boldsymbol{\theta}^\star) \ni \boldsymbol{\theta}$.*

**Lemma A.14.** *(Fenchel-Young inequality) (Fenchel, 1949) For any $\psi : \mathbb{R}^p \to \mathbb{R}$ and $\boldsymbol{\omega}, \boldsymbol{\theta} \in \mathbb{R}^p$, it holds:*

$$\langle \boldsymbol{\theta}, \boldsymbol{\omega} \rangle \leq \psi(\boldsymbol{\theta}) + \psi^\star(\boldsymbol{\omega}). \tag{45}$$

Next, we show that the scaled version of the iterates from Lemma A.10, $\widetilde{\boldsymbol{\theta}}_t$, is an $\left(D_{\frac{1}{2}\|\cdot\|_\star^2}^{\widetilde{\boldsymbol{\theta}}_t}, \epsilon(t), \delta(t)\right)$-approximate KKT point for $\epsilon(t)$ and $\delta(t)$ that vanish as $t$ increases.

**Proposition A.15.** *For any $t > t_0$, $\widetilde{\boldsymbol{\theta}}_t = \dfrac{\boldsymbol{\theta}_t}{\left(\min_{i \in [m]} y_i f(\mathbf{x}_i; \boldsymbol{\theta}_t)\right)^{\frac{1}{L}}}$ is an $\left(D_{\frac{1}{2}\|\cdot\|_\star^2}^{\widetilde{\boldsymbol{\theta}}_t}, \epsilon(t), \delta(t)\right)$-approximate KKT point of Equation 40, with:*

$$\begin{aligned}
\epsilon(t) &= \frac{1}{\widetilde{\gamma}(t_0)^{\frac{2}{L}}} \left(1 - \left\langle \frac{\boldsymbol{\theta}_t}{\|\boldsymbol{\theta}_t\|}, \frac{-\mathbf{g}_t^\star}{\|\mathbf{g}_t^\star\|_\star} \right\rangle \right), \\
\delta(t) &= \frac{m}{eL\widetilde{\gamma}(t_0)^{\frac{2}{L}} \log \frac{1}{\mathcal{L}}},
\end{aligned} \tag{46}$$

*with $\mathbf{g}_t^\star \in \arg\min_{\mathbf{u} \in \partial \mathcal{L}(\boldsymbol{\theta}_t)} \|\mathbf{u}\|_\star$.*

*Proof.* We suppress the dependence of the loss and the iterates from the time index $t$, when it is obvious from the context. From Lemma A.10, we know that $\widetilde{\boldsymbol{\theta}}$ is a feasible point. To simplify the notation, let $q_{\min} = \min_{i \in [m]} y_i f(\mathbf{x}_i; \boldsymbol{\theta})$. We will denote by $\widetilde{\mathbf{k}} \in \partial \frac{1}{2}\|\widetilde{\boldsymbol{\theta}}\|^2$ any subgradient of $\frac{1}{2}\|\cdot\|^2$ at $\widetilde{\boldsymbol{\theta}}$. Let, as previously stated, $\mathbf{g}_t^\star \in \arg\min_{\mathbf{u} \in \partial \mathcal{L}(\boldsymbol{\theta}_t)} \|\mathbf{u}\|_\star$ and $\mathbf{h}_i^\star \in \partial f(\mathbf{x}_i; \boldsymbol{\theta}), i \in [m]$, such that $\mathbf{g}_t^\star = -\sum_{i=1}^m e^{-y_i f(\mathbf{x}_i; \boldsymbol{\theta})} y_i \mathbf{h}_i^\star$ (whose existence is guaranteed from chain rule - Theorem A.1). Finally, we define $\widetilde{\mathbf{h}}_i^\star = q_{\min}^{\frac{1}{L}-1} \mathbf{h}_i^\star$ for all $i \in [m]$, for which it holds: $\widetilde{\mathbf{h}}_i^\star \in \partial f(\mathbf{x}_i; \widetilde{\boldsymbol{\theta}})$ from Theorem B.2(a) in Lyu & Li (2020).

Given all these definitions, we set $\lambda_i = \frac{\|\boldsymbol{\theta}\|}{\|\mathbf{g}_t^\star\|_\star} q_{\min}^{1-\frac{2}{L}} e^{-y_i f(\mathbf{x}_i; \boldsymbol{\theta})} \geq 0$. The dual vector from the $(d, \epsilon, \delta)$-stationarity definition can be simplified to:

$$\begin{aligned}
\sum_{i=1}^m \lambda_i y_i \widetilde{\mathbf{h}}_i^\star &= \sum_{i=1}^m \lambda_i q_{\min}^{\frac{1}{L}-1} y_i \mathbf{h}_i^\star \quad \text{(Thm B.2(a) in Lyu \& Li (2020))} \\
&= \frac{\|\boldsymbol{\theta}\|}{q_{\min}^{\frac{1}{L}}\|\mathbf{g}_t^\star\|_\star} \sum_{i=1}^m e^{-y_i f(\mathbf{x}_i; \boldsymbol{\theta})} y_i \mathbf{h}_i \\
&= -\frac{\|\boldsymbol{\theta}\|\mathbf{g}_t^\star}{q_{\min}^{\frac{1}{L}}\|\mathbf{g}_t^\star\|_\star},
\end{aligned} \tag{47}$$

which is a scaled version of the (minimum norm) subderivative of the loss.

Let $\psi(\boldsymbol{\theta}) = \frac{1}{2}\|\boldsymbol{\theta}\|_\star^2$ be the potential function that we shall use in order to define our divergence. For this specific $\psi$, it holds: $\psi^\star(\boldsymbol{\omega}) = \frac{1}{2}\|\boldsymbol{\omega}\|^2$ (see for instance Example 3.27 in Boyd & Vandenberghe (2014) for a derivation). Recall that in the definition of $D^{\mathbf{m}}_{\frac{1}{2}\|\cdot\|_\star^2}$ (Equation 12) there is an extra choice that we have to make; the one of the subderivative $\mathbf{m}$. In what follows, we will specifically measure "distance" between $\sum_{i=1}^m \lambda_i y_i \widetilde{\mathbf{h}}_i$ and $\widetilde{\mathbf{k}}$ using $D^{\widetilde{\boldsymbol{\theta}}}_{\frac{1}{2}\|\cdot\|_\star^2}(\cdot, \cdot)$, i.e. by picking $\mathbf{m} = \widetilde{\boldsymbol{\theta}}$. This is possible, since from Proposition A.13 it holds that $\widetilde{\boldsymbol{\theta}} \in \partial \frac{1}{2}\|\widetilde{\mathbf{k}}\|_\star^2$. Finally, let $\mathbf{r} \in \partial\|\widetilde{\boldsymbol{\theta}}\|$ be the subgradient of $\|\cdot\|$ that stems from the chain rule of $\frac{1}{2}\|\cdot\|^2$ evaluated at $\widetilde{\boldsymbol{\theta}}$. We calculate the divergence between the two vectors:

$$
D^{\widetilde{\boldsymbol{\theta}}}_{\frac{1}{2}\|\cdot\|_\star^2}\left(\sum_{i=1}^m \lambda_i y_i \widetilde{\mathbf{h}}_i, \widetilde{\mathbf{k}}\right)
$$

$$
= \frac{1}{2}\left\|-\frac{\|\boldsymbol{\theta}\|\mathbf{g}_t^\star}{q_{\min}^{\frac{1}{L}}\|\mathbf{g}_t^\star\|_\star}\right\|_\star^2 - \frac{1}{2}\left\|\widetilde{\mathbf{k}}\right\|_\star^2 - \left\langle \frac{\boldsymbol{\theta}}{q_{\min}^{\frac{1}{L}}}, -\frac{\|\boldsymbol{\theta}\|\mathbf{g}_t^\star}{q_{\min}^{\frac{1}{L}}\|\mathbf{g}_t^\star\|_\star} - \widetilde{\mathbf{k}}\right\rangle
$$

$$
= \frac{1}{2}\frac{\|\boldsymbol{\theta}\|^2}{q_{\min}^{\frac{2}{L}}} - \frac{1}{2}\left\|\|\widetilde{\boldsymbol{\theta}}\|\mathbf{r}\right\|_\star^2 - \left\langle \frac{\boldsymbol{\theta}}{q_{\min}^{\frac{1}{L}}}, \frac{-\|\boldsymbol{\theta}\|\mathbf{g}_t^\star}{q_{\min}^{\frac{1}{L}}\|\mathbf{g}_t^\star\|_\star} - \|\widetilde{\boldsymbol{\theta}}\|\mathbf{r}\right\rangle \quad \text{(Chain rule)}
$$

$$
= \frac{\|\boldsymbol{\theta}\|^2}{q_{\min}^{\frac{2}{L}}}\left(\frac{1}{2} - \frac{1}{2}\|\mathbf{r}\|_\star^2 - \left\langle\frac{\boldsymbol{\theta}}{\|\boldsymbol{\theta}\|}, \frac{-\mathbf{g}_t^\star}{\|\mathbf{g}_t^\star\|_\star}\right\rangle + \left\langle\frac{\boldsymbol{\theta}}{\|\boldsymbol{\theta}\|}, \mathbf{r}\right\rangle\right)
$$

$$
\leq \frac{\|\boldsymbol{\theta}\|^2}{q_{\min}^{\frac{2}{L}}}\left(\frac{1}{2} - \frac{1}{2}\|\mathbf{r}\|_\star^2 - \left\langle\frac{\boldsymbol{\theta}}{\|\boldsymbol{\theta}\|}, \frac{-\mathbf{g}_t^\star}{\|\mathbf{g}_t^\star\|_\star}\right\rangle + \frac{1}{2}\left\|\frac{\boldsymbol{\theta}}{\|\boldsymbol{\theta}\|}\right\|^2 + \frac{1}{2}\|\mathbf{r}\|_\star^2\right) \quad \text{(Equation 45)}
$$

$$
= \frac{\|\boldsymbol{\theta}\|^2}{q_{\min}^{\frac{2}{L}}}\left(1 - \left\langle\frac{\boldsymbol{\theta}}{\|\boldsymbol{\theta}\|}, \frac{-\mathbf{g}_t^\star}{\|\mathbf{g}_t^\star\|_\star}\right\rangle\right)
$$

$$
\leq \frac{1}{\widetilde{\gamma}^{\frac{2}{L}}}\left(1 - \left\langle\frac{\boldsymbol{\theta}}{\|\boldsymbol{\theta}\|}, \frac{-\mathbf{g}_t^\star}{\|\mathbf{g}_t^\star\|_\star}\right\rangle\right) \leq \frac{1}{\widetilde{\gamma}(t_0)^{\frac{2}{L}}}\left(1 - \left\langle\frac{\boldsymbol{\theta}}{\|\boldsymbol{\theta}\|}, \frac{-\mathbf{g}_t^\star}{\|\mathbf{g}_t^\star\|_\star}\right\rangle\right),
$$

(48)

where the last 2 inequalities follow from the relation between smooth and hard margin (Lemma A.7), and the monotonicity of the former. For the second condition of an approximate KKT point, we have:

$$
\sum_{i=1}^m \lambda_i\left(y_i f(\mathbf{x}_i; \widetilde{\boldsymbol{\theta}}) - 1\right) = \frac{\|\boldsymbol{\theta}\|}{\|\mathbf{g}_t^\star\|_\star}\sum_{i=1}^m q_{\min}^{1-\frac{2}{L}} e^{-y_i f(\mathbf{x}_i; \boldsymbol{\theta})}\left(\frac{y_i f(\mathbf{x}_i; \boldsymbol{\theta})}{q_{\min}} - 1\right)
$$

$$
= \frac{\|\boldsymbol{\theta}\|}{q_{\min}^{\frac{2}{L}}\|\mathbf{g}_t^\star\|_\star}\sum_{i=1}^m e^{-y_i f(\mathbf{x}_i; \boldsymbol{\theta})}\left(y_i f(\mathbf{x}_i; \boldsymbol{\theta}) - q_{\min}\right).
$$

(49)

From eq. Equation 30 and Equation 28, we can lower bound the dual norm of the subderivate:

$$
\|\mathbf{g}_t^\star\|_\star \geq \frac{L}{\|\boldsymbol{\theta}\|}\mathcal{L}\log\frac{1}{\mathcal{L}} \geq \frac{L}{\|\boldsymbol{\theta}\|}e^{-q_{\min}}\log\frac{1}{\mathcal{L}}.
$$

(50)

By plugging in back to eq. Equation 49, we obtain

$$
\sum_{i=1}^m \lambda_i\left(y_i f(\mathbf{x}_i; \widetilde{\boldsymbol{\theta}}) - 1\right) \leq \frac{\|\boldsymbol{\theta}\|^2}{q_{\min}^{\frac{2}{L}}Le^{-q_{\min}}\log\frac{1}{\mathcal{L}}}\sum_{i=1}^m e^{-y_i f(\mathbf{x}_i; \boldsymbol{\theta})}\left(y_i f(\mathbf{x}_i; \boldsymbol{\theta}) - q_{\min}\right)
$$

$$
= \frac{\|\boldsymbol{\theta}\|^2}{q_{\min}^{\frac{2}{L}}L\log\frac{1}{\mathcal{L}}}\sum_{i=1}^m e^{-(y_i f(\mathbf{x}_i; \boldsymbol{\theta}) - q_{\min})}\left(y_i f(\mathbf{x}_i; \boldsymbol{\theta}) - q_{\min}\right)
$$

(51)

$$
\leq \frac{1}{\widetilde{\gamma}(t_0)^{\frac{2}{L}}L\log\frac{1}{\mathcal{L}}}\sum_{i=1}^m e^{-(y_i f(\mathbf{x}_i; \boldsymbol{\theta}) - q_{\min})}\left(y_i f(\mathbf{x}_i; \boldsymbol{\theta}) - q_{\min}\right) \quad \text{(Lemmata A.7, A.5)}
$$

$$
\leq \frac{m}{e\widetilde{\gamma}(t_0)^{\frac{2}{L}}L\log\frac{1}{\mathcal{L}}},
$$

since the function $u \mapsto e^{-u}u, u > 0$ has a maximum value of $e^{-1}$. $\qquad\square$

Before we proceed with the main result, we state and prove two useful Lemmata. The first one lower bounds the alignment between normalized iterates and normalized loss gradients. This Lemma is key for showing that the alignment goes to 1 as $t \to \infty$.

**Lemma A.16.** *For all $t_2 > t_1 \geq t_0$, there exists $t_\star \in (t_1, t_2)$ such that:*

$$\left( \frac{1}{\left\langle \frac{\boldsymbol{\theta}_{t_\star}}{\|\boldsymbol{\theta}_{t_\star}\|}, \frac{-\mathbf{g}_{t_\star}^\star}{\|\mathbf{g}_{t_\star}^\star\|_\star} \right\rangle} - 1 \right) \leq \frac{1}{L} \frac{\log \frac{\widetilde{\gamma}(t_2)}{\widetilde{\gamma}(t_1)}}{\int_{t_1}^{t_2} \frac{\left\| \frac{d\boldsymbol{\theta}_t}{dt} \right\|}{\|\boldsymbol{\theta}_t\|} dt}, \tag{52}$$

*for all $\mathbf{g}_{t_\star}^\star \in \arg\min_{\mathbf{u} \in \partial \mathcal{L}(\boldsymbol{\theta}_{t_\star})} \|\mathbf{u}\|_\star$*

*Proof.* From Lemma A.5, we have for all $\mathbf{g}_t^\star \in \arg\min_{\mathbf{u} \in \partial \mathcal{L}(\boldsymbol{\theta}_t)} \|\mathbf{u}\|_\star$:

$$\begin{aligned} \frac{d\log\widetilde{\gamma}}{dt} &\geq L \left\| \frac{d\boldsymbol{\theta}_t}{dt} \right\|^2 \left( \frac{1}{\langle \boldsymbol{\theta}_t, -\mathbf{g}_t^\star \rangle} - \frac{1}{\|\boldsymbol{\theta}_t\| \left\| \frac{d\boldsymbol{\theta}_t}{dt} \right\|} \right) \\ &= L \frac{\left\| \frac{d\boldsymbol{\theta}_t}{dt} \right\|}{\|\boldsymbol{\theta}_t\|} \left( \frac{1}{\left\langle \frac{\boldsymbol{\theta}_t}{\|\boldsymbol{\theta}_t\|}, \frac{-\mathbf{g}_t^\star(\boldsymbol{\theta}_t)}{\|\mathbf{g}_t^\star\|_\star} \right\rangle} - 1 \right). \end{aligned} \tag{53}$$

We then integrate the two sides from $t_1$ to $t_2 > t_1 > t_0$:

$$\int_{t_1}^{t_2} \left( \frac{1}{\left\langle \frac{\boldsymbol{\theta}_t}{\|\boldsymbol{\theta}_t\|}, \frac{-\mathbf{g}_t^\star}{\|\mathbf{g}_t^\star\|_\star} \right\rangle} - 1 \right) \frac{\left\| \frac{d\boldsymbol{\theta}_t}{dt} \right\|}{\|\boldsymbol{\theta}_t\|} dt \leq \frac{1}{L} \log \frac{\widetilde{\gamma}(t_2)}{\widetilde{\gamma}(t_1)}. \tag{54}$$

The desired existential statement follows from a proof by contradiction. $\qquad\square$

Next, we bound the rate of change of the normalized iterates.

**Lemma A.17.** *For any $t > 0$, it holds:*

$$\left\| \frac{d\frac{\boldsymbol{\theta}_t}{\|\boldsymbol{\theta}_t\|}}{dt} \right\| \leq 2 \frac{\left\| \frac{d\boldsymbol{\theta}_t}{dt} \right\|}{\|\boldsymbol{\theta}_t\|}. \tag{55}$$

*Proof.* The rate of change of the normalized iterates can be written as follows:

$$\begin{aligned} \frac{d\frac{\boldsymbol{\theta}_t}{\|\boldsymbol{\theta}_t\|}}{dt} &= \frac{1}{\|\boldsymbol{\theta}_t\|} \frac{d\boldsymbol{\theta}_t}{dt} + \boldsymbol{\theta}_t \left( -\frac{1}{\|\boldsymbol{\theta}_t\|^2} \frac{d\|\boldsymbol{\theta}_t\|}{dt} \right) \\ &= \frac{1}{\|\boldsymbol{\theta}_t\|} \frac{d\boldsymbol{\theta}_t}{dt} + \boldsymbol{\theta}_t \left( -\frac{1}{\|\boldsymbol{\theta}_t\|^2} \left\langle \mathbf{n}_t, \frac{d\boldsymbol{\theta}_t}{dt} \right\rangle \right), \quad \text{(Chain rule)} \end{aligned} \tag{56}$$

where $\mathbf{n}_t \in \partial\|\boldsymbol{\theta}_t\|$. So, by the triangle inequality, its norm is bounded by:

$$\begin{aligned} \left\| \frac{d\frac{\boldsymbol{\theta}_t}{\|\boldsymbol{\theta}_t\|}}{dt} \right\| &\leq \frac{\left\| \frac{d\boldsymbol{\theta}_t}{dt} \right\|}{\|\boldsymbol{\theta}_t\|} + \frac{1}{\|\boldsymbol{\theta}_t\|} \left| \left\langle \mathbf{n}_t, \frac{d\boldsymbol{\theta}_t}{dt} \right\rangle \right| \\ &\leq 2 \frac{\left\| \frac{d\boldsymbol{\theta}_t}{dt} \right\|}{\|\boldsymbol{\theta}_t\|}. \quad \text{(definition of dual norm and } \|\mathbf{n}_t\|_\star \leq 1) \end{aligned} \tag{57}$$

$\qquad\square$

We are, now, ready to state and prove our main result.

**Theorem A.18.** *For steepest flow (eq. Equation 3) on the exponential loss, under assumptions A1, A2, A3, any limit point $\bar{\boldsymbol{\theta}}$ of $\left\{ \frac{\boldsymbol{\theta}_t}{\|\boldsymbol{\theta}_t\|} \right\}_{t \geq 0}$ is along the direction of a $D^{\widetilde{\boldsymbol{\theta}}}_{\frac{1}{2}\|\cdot\|^2_\star}$-generalized KKT point, $\widetilde{\boldsymbol{\theta}} := \frac{\bar{\boldsymbol{\theta}}}{\left( \min_{i \in [m]} y_i f(\mathbf{x}_i; \bar{\boldsymbol{\theta}}) \right)^{\frac{1}{L}}}$, of the following optimization problem:*

$$
\min_{\boldsymbol{\theta}} \frac{1}{2} \|\boldsymbol{\theta}\|^2 \tag{58}
$$
$$
\text{s.t. } y_i f(\mathbf{x}_i; \boldsymbol{\theta}) \geq 1, \ \forall i \in [m].
$$

*Proof.* Our strategy will be to consider any limit point $\bar{\boldsymbol{\theta}}$ and construct $\left( D^{\widetilde{\boldsymbol{\theta}}_t}_{\frac{1}{2}\|\cdot\|^2_\star}, \epsilon(t), \delta(t) \right)$-approximate KKT points that converge to it, with vanishing $\epsilon(t), \delta(t)$.

Let $\epsilon_m = \frac{1}{m}$ for any $m > 0$. We construct a sequence $\{t_m\}_{m \geq 0}$, by induction, in the following sense. Suppose $t_1 < \ldots < t_{m-1}$ have been constructed already. Since $\bar{\boldsymbol{\theta}}$ is a limit point of the normalized iterates and $\log \widetilde{\gamma}_t \to \log \widetilde{\gamma}_\infty < \infty$ (as $\widetilde{\gamma}_t$ is non-decreasing and bounded from above), there exists $s_m > t_{m-1}$ such that:

$$
\left\| \frac{\boldsymbol{\theta}_{s_m}}{\|\boldsymbol{\theta}_{s_m}\|} - \bar{\boldsymbol{\theta}} \right\| \leq \epsilon_m = \frac{1}{m} \quad \text{and} \quad \frac{1}{L} \log \frac{\widetilde{\gamma}_\infty}{\widetilde{\gamma}_{s_m}} \leq \epsilon_m^2 = \frac{1}{m^2}. \tag{59}
$$

Since $\frac{d \log \|\boldsymbol{\theta}_t\|}{dt} \leq \frac{\left\| \frac{d\boldsymbol{\theta}_t}{dt} \right\|}{\|\boldsymbol{\theta}_t\|}$, we have that $\lim_{t \to \infty} \int_{t_A}^t \frac{\left\| \frac{d\boldsymbol{\theta}_{t'}}{dt'} \right\|}{\|\boldsymbol{\theta}_{t'}\|} dt' = \infty$ for all $t_A > 0$. Thus, there exists $s'_m > s_m$ such that $\int_{s_m}^{s'_m} \frac{\left\| \frac{d\boldsymbol{\theta}_t}{dt} \right\|}{\|\boldsymbol{\theta}_t\|} dt = \frac{1}{m}$. Now, from Lemma A.16, we know there exists $t_\star \in (s_m, s'_m)$ with:

$$
\left( \frac{1}{\left\langle \frac{\boldsymbol{\theta}_{t_\star}}{\|\boldsymbol{\theta}_{t_\star}\|}, \frac{-\mathbf{g}_t^\star}{\|\mathbf{g}_t^\star\|_\star} \right\rangle} - 1 \right) \leq \frac{1}{L} \frac{\log \frac{\widetilde{\gamma}_{s'_m}}{\widetilde{\gamma}_{s_m}}}{\int_{s_m}^{s'_m} \frac{\left\| \frac{d\boldsymbol{\theta}_t}{dt} \right\|}{\|\boldsymbol{\theta}_t\|} dt} \leq \frac{\frac{1}{m^2}}{\frac{1}{m}} = \frac{1}{m}, \tag{60}
$$

which implies $\left\langle \frac{\boldsymbol{\theta}_{t_\star}}{\|\boldsymbol{\theta}_{t_\star}\|}, \frac{-\mathbf{g}_t^\star}{\|\mathbf{g}_t^\star\|_\star} \right\rangle \geq \frac{1}{1 + \frac{1}{m}} \to 1$ as $m \to \infty$. On the other hand, for the normalized iterates we have:

$$
\left\| \frac{\boldsymbol{\theta}_{t_\star}}{\|\boldsymbol{\theta}_{t_\star}\|} - \bar{\boldsymbol{\theta}} \right\| \leq \left\| \frac{\boldsymbol{\theta}_{t_\star}}{\|\boldsymbol{\theta}_{t_\star}\|} - \frac{\boldsymbol{\theta}_{s_m}}{\|\boldsymbol{\theta}_{s_m}\|} \right\| + \left\| \frac{\boldsymbol{\theta}_{s_m}}{\|\boldsymbol{\theta}_{s_m}\|} - \bar{\boldsymbol{\theta}} \right\| \overset{\text{eq. Equation 59}}{\leq} \left\| \frac{\boldsymbol{\theta}_{t_\star}}{\|\boldsymbol{\theta}_{t_\star}\|} - \frac{\boldsymbol{\theta}_{s_m}}{\|\boldsymbol{\theta}_{s_m}\|} \right\| + \frac{1}{m}. \tag{61}
$$

To deal with the first term, we can leverage Lemma A.17 which bounds the rate of change of the normalized iterates:

$$
\left\| \frac{\boldsymbol{\theta}_{t_\star}}{\|\boldsymbol{\theta}_{t_\star}\|} - \bar{\boldsymbol{\theta}} \right\| \leq 2 \int_{s_m}^{t_\star} \frac{\left\| \frac{d\boldsymbol{\theta}_t}{dt} \right\|}{\|\boldsymbol{\theta}_t\|} dt + \frac{1}{m} \leq 2 \int_{s_m}^{s'_m} \frac{\left\| \frac{d\boldsymbol{\theta}_t}{dt} \right\|}{\|\boldsymbol{\theta}_t\|} dt + \frac{1}{m} = \frac{3}{m} \to 0. \tag{62}
$$

Hence, by picking $t_m$ as $t_\star$, we constructed a time sequence such that, for any limit point $\bar{\boldsymbol{\theta}}$, $\frac{\boldsymbol{\theta}_{t_m}}{\|\boldsymbol{\theta}_{t_m}\|} \to \bar{\boldsymbol{\theta}}$ and also $\left\langle \frac{\boldsymbol{\theta}_{t_m}}{\|\boldsymbol{\theta}_{t_m}\|}, \frac{-\mathbf{g}_t^\star(\boldsymbol{\theta}_{t_m})}{\|\mathbf{g}_t^\star(\boldsymbol{\theta}_{t_m})\|_\star} \right\rangle \to 1$.

Then, from Proposition A.15, we know that $\widetilde{\boldsymbol{\theta}}_{t_m} = \frac{\boldsymbol{\theta}_{t_m}}{\left( \min_{i \in [m]} y_i f(\mathbf{x}_i; \boldsymbol{\theta}_{t_m}) \right)^{\frac{1}{L}}}$ is an $\left( D^{\widetilde{\boldsymbol{\theta}}_t}_{\frac{1}{2}\|\cdot\|^2_\star}, \epsilon(t_m), \delta(t_m) \right)$-approximate KKT point of Equation 40. But $\epsilon(t_m) \to 0$ (since the alignment goes to 1) and $\delta(t_m) \to 0$ as the loss goes to zero (Lemma A.6), thus the sequence satisfies the conditions of Proposition A.20, which shows that the limit point of the sequence is a generalized KKT point. This concludes the proof of our claim. $\square$

While the previous result is not strong enough to guarantee convergence to an approximate KKT point for a general algorithm norm $\|\cdot\|$, it immediately implies it in the case of a smooth norm. The proof relies on a fundamental relationship between smoothness of a function and strong convexity of its convex conjugate.

**Proposition A.19.** *(Conjugate Correspondence Theorem - Thm. 5.26 in Beck (2017)) Let $\sigma > 0$. If $\psi$ is a $\frac{1}{\sigma}$-smooth convex function, then its conjugate $\psi^\star$ is $\sigma$-strongly convex.*

We can prove the following corollary for a special class of steepest flows.

**Corollary A.19.1.** *For steepest flow (eq. Equation 3) with respect to a norm $\|\cdot\|$, whose square is a smooth function, on the exponential loss, under assumptions A1, A2, A3, any limit point $\bar{\boldsymbol{\theta}}$ of $\left\{ \frac{\boldsymbol{\theta}_t}{\|\boldsymbol{\theta}_t\|} \right\}_{t \geq 0}$ is along the direction of a KKT point of optimization problem Equation 40.*

*Proof.* From Proposition A.19, if $\frac{1}{2}\|\cdot\|^2$ is $\frac{1}{\sigma}$-smooth w.r.t. $\|\cdot\|$, then the function $\frac{1}{2}\|\cdot\|_\star^2$ is $\sigma$-strongly convex w.r.t. $\|\cdot\|_\star$. Thus, the function $D_{\frac{1}{2}\|\cdot\|_\star^2}^{\boldsymbol{\theta}}$ is defined with respect to a strongly convex function and it becomes a proper Bregman divergence. Hence, from Theorem 5.24 in Beck (2017), for $\widetilde{\mathbf{h}}_i^\star = q_{\min}^{\frac{1}{L}-1} \mathbf{h}_i^\star$, where $\widetilde{\mathbf{h}}_i^\star \in \partial f(\mathbf{x}_i; \widetilde{\boldsymbol{\theta}}), i \in [m]$ such that $\mathbf{g}_t^\star = -\sum_{i=1}^m e^{-y_i f(\mathbf{x}_i;\boldsymbol{\theta})} y_i \mathbf{h}_i^\star$ and $\widetilde{\mathbf{k}} \in \partial \frac{1}{2}\|\widetilde{\boldsymbol{\theta}}\|^2$, it holds:

$$D_{\frac{1}{2}\|\cdot\|_\star^2}^{\widetilde{\boldsymbol{\theta}}} \left( \sum_{i=1}^m \lambda_i y_i \widetilde{\mathbf{h}}_i^\star, \widetilde{\mathbf{k}} \right) \geq \sigma \left\| \sum_{i=1}^m \lambda_i y_i \widetilde{\mathbf{h}}_i^\star - \widetilde{\mathbf{k}} \right\|_\star. \tag{63}$$

In other words, if $D_{\frac{1}{2}\|\cdot\|_\star^2}^{\widetilde{\boldsymbol{\theta}}}(\boldsymbol{\alpha}, \boldsymbol{\beta})$ is 0, so is the difference $\boldsymbol{\alpha} - \boldsymbol{b}$ for any $\boldsymbol{\alpha}, \boldsymbol{b}$. As a result, and from the equivalence of the norms, the sequence $\widetilde{\boldsymbol{\theta}}_{t_m} = \frac{\boldsymbol{\theta}_{t_m}}{\left( \min_{i \in [m]} y_i f(\mathbf{x}_i;\boldsymbol{\theta}_{t_m}) \right)^{\frac{1}{L}}}$ from the proof of Theorem A.18, induces a sequence of $(\epsilon(t_m), \delta(t_m))$-approximate KKT points, which converges to a KKT point of Equation 40. By Theorem C.4 in Lyu & Li (2020) (which is itself based on a result due to Dutta et al. (2013)), we get that $\frac{\bar{\boldsymbol{\theta}}}{\left( \min_{i \in [m]} y_i f(\mathbf{x}_i;\boldsymbol{\theta}_t) \right)^{\frac{1}{L}}}$ is a KKT point of Equation 40. $\qquad \square$

The following technical result was used in the proof of Theorem A.18.

**Proposition A.20.** *Let (MM) be the following optimization problem:*

$$\min_{\boldsymbol{\theta} \in \mathbb{R}^d} \frac{1}{2}\|\boldsymbol{\theta}\|^2 \tag{MM}$$
$$s.t. \ y_i f(\mathbf{x}_i; \boldsymbol{\theta}) \geq 1, \ \forall i \in [m].$$

*Let $\{\boldsymbol{\theta}_t\}_{t \geq 0}$ be a sequence of feasible, $(d, \epsilon_t, \delta_t)$-approximate KKT points, with $d := D_{\frac{1}{2}\|\cdot\|_\star^2}^{\boldsymbol{\theta}_t}$ with $\epsilon_t \downarrow 0, \delta_t \downarrow 0$ and $\boldsymbol{\theta}_t \to \bar{\boldsymbol{\theta}}$. Assume that the Mangasarian-Fromovitz- Constraint Qualifications hold at $\bar{\boldsymbol{\theta}}$. Then, $\bar{\boldsymbol{\theta}}$ is a $D_{\frac{1}{2}\|\cdot\|_\star^2}^{\bar{\boldsymbol{\theta}}}$-generalized KKT point of (MM).*

*Proof.* Our proof closely follows the proof of Theorem 3.6 in (Dutta et al., 2013), which is the direct analog for $(\epsilon, \delta)$-approximate KKT points.

By the definition of the $(d, \epsilon_t, \delta_t)$ stationarity, for each $t > 0$, there exist $\mathbf{h}_i^t \in \partial f(\mathbf{x}_i; \boldsymbol{\theta}_t), i \in [m], \mathbf{k}^t \in \partial \frac{1}{2}\|\boldsymbol{\theta}_t\|^2$ and $\lambda_i^t \geq 0, i \in [m]$ such that:

(i) $D_{\frac{1}{2}\|\cdot\|_\star^2}^{\boldsymbol{\theta}_t} \left( \sum_{i=1}^m \lambda_i^t y_i \mathbf{h}_i^t, \mathbf{k}^t \right) \leq \epsilon_t.$

(ii) $\sum_{i=1}^m \lambda_i^t \left( y_i f(\mathbf{x}_i; \boldsymbol{\theta}_t) - 1 \right) \leq \delta_t.$

We will show that the sequence $\{\boldsymbol{\lambda}^t\}_{t \geq 0}$ is bounded. Assume on the contrary that is not. Consider $\hat{\boldsymbol{\lambda}} = \frac{\boldsymbol{\lambda}^t}{\|\boldsymbol{\lambda}^t\|}$, which is bounded and wlog, it is: $\hat{\boldsymbol{\lambda}} \to \widetilde{\boldsymbol{\lambda}}$ with $\|\hat{\boldsymbol{\lambda}}\| = 1$. Note that the sequences $\{\mathbf{h}_i^t\}, \{\mathbf{k}^t\}$ are bounded, as elements of a subdifferential, by the Lipschitz constant of the corresponding function. Hence, they also converge to, say, $\bar{\mathbf{h}}_i$ for all $i \in [m]$ and $\bar{\mathbf{k}}$, respectively. From

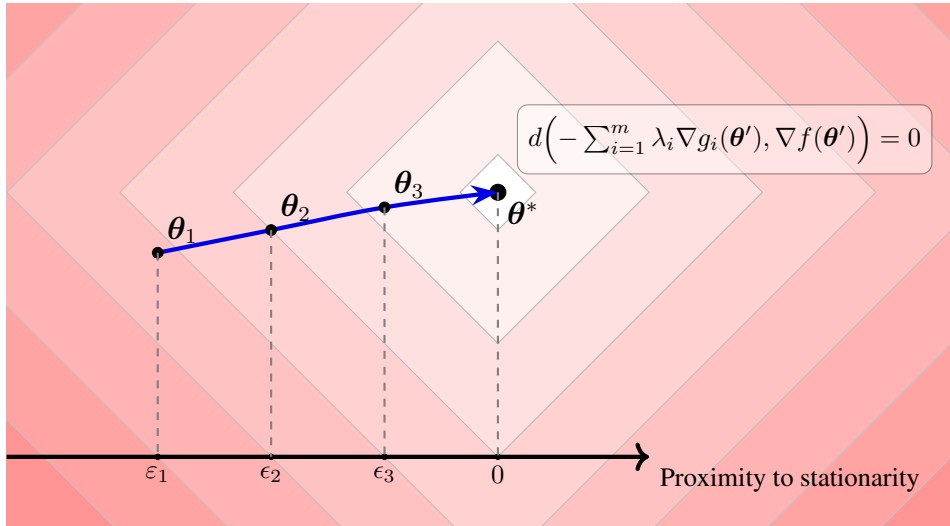

Figure 3: An illustration of the reduction of the Bregman proximity measure. Once it separates the training points, steepest descent in homogeneous networks implicitly makes progress towards generalized stationary points $\boldsymbol{\theta}^\star$ of a margin maximization problem (Theorem A.18).

condition (i), we have:

$$\frac{1}{\|\boldsymbol{\lambda}^t\|^2} D_{\frac{1}{2}\|\cdot\|_\star^2}^{\boldsymbol{\theta}_t}\left(\sum_{i=1}^m \lambda_i^t y_i \mathbf{h}_i^t, \mathbf{k}^t\right) \leq \frac{\epsilon_t}{\|\boldsymbol{\lambda}_t\|^2} \iff$$

$$\frac{1}{2}\left\|\sum_{i=1}^m \frac{\lambda_i^t}{\|\boldsymbol{\lambda}^t\|} y_i \mathbf{h}_i^t\right\|_\star^2 - \frac{1}{2}\left\|\frac{\mathbf{k}^t}{\|\boldsymbol{\lambda}^t\|}\right\|_\star^2 - \left\langle\frac{\boldsymbol{\theta}_t}{\|\boldsymbol{\lambda}^t\|}, \sum_{i=1}^m \frac{\lambda_i^t}{\|\boldsymbol{\lambda}\|} y_i \mathbf{h}_i^t - \frac{\mathbf{k}^t}{\|\boldsymbol{\lambda}\|}\right\rangle \leq \frac{\epsilon_t}{\|\boldsymbol{\lambda}_t\|^2} \tag{64}$$

hence, by taking $t \to \infty$, we get:

$$\frac{1}{2}\left\|\sum_{i=1}^m \widetilde{\lambda}_i y_i \bar{\mathbf{h}}_i\right\|_\star^2 \leq 0, \tag{65}$$

which implies that there exists $\widetilde{\boldsymbol{\lambda}} \neq \mathbf{0}$ such that $\sum_{i=1}^m \widetilde{\lambda}_i y_i \bar{\mathbf{h}}_i = \mathbf{0}$, where recall $\bar{\mathbf{h}}_i \in \partial f(\mathbf{x}_i; \bar{\boldsymbol{\theta}})$. An existence of such a vector is prohibited by the Mangasarian-Fromovitz-Constraint Qualifications which hold at $\bar{\boldsymbol{\theta}}$. Thus, $\{\boldsymbol{\lambda}^t\}_{t\geq 0}$ is bounded and $\boldsymbol{\lambda}^t \to \bar{\boldsymbol{\lambda}}$ for some $\bar{\boldsymbol{\lambda}} \in \mathbb{R}^m$. Hence, by taking the limit $t \to \infty$, we have:

$$D_{\frac{1}{2}\|\cdot\|_\star^2}^{\bar{\boldsymbol{\theta}}}\left(\sum_{i=1}^m \bar{\lambda}_i y_i \bar{\mathbf{h}}_i, \bar{\mathbf{k}}\right) \leq 0, \tag{66}$$

where $\bar{\mathbf{h}}_i \in \partial f(\mathbf{x}_i; \bar{\boldsymbol{\theta}}), i \in [m]$ and $\bar{\mathbf{k}} \in \partial \frac{1}{2}\|\bar{\boldsymbol{\theta}}\|$. From (ii), we obtain $\sum_{i=1}^m \bar{\lambda}_i \left(y_i f(\mathbf{x}_i; \bar{\boldsymbol{\theta}}) - 1\right) \leq 0$. However, $y_i f(\mathbf{x}_i; \bar{\boldsymbol{\theta}}) - 1 \geq 0$ ($\bar{\boldsymbol{\theta}}$ is a feasible point of (P)) and $\bar{\lambda}_i \geq 0$ for all $i \in [m]$, thus it holds:

$$\sum_{i=1}^m \bar{\lambda}_i \left(y_i f(\mathbf{x}_i; \bar{\boldsymbol{\theta}}) - 1\right) = 0, \tag{67}$$

which concludes the proof that $\bar{\boldsymbol{\theta}}$ is a $D_{\frac{1}{2}\|\cdot\|_\star^2}^{\bar{\boldsymbol{\theta}}}$-generalized KKT point.

$\square$

### A.3 GENERALIZATION TO OTHER LOSSES

The previous results can be generalized to any loss with exponential tails. In particular, let us proceed to the following definition:

**Definition A.21.** *Let $\Phi : \mathbb{R} \to \mathbb{R}$. Assume that $\mathcal{L}(\boldsymbol{\theta}) = \sum_{i=1}^{m} e^{-\Phi(y_i f(\mathbf{x}_i;\boldsymbol{\theta}))}, \boldsymbol{\theta} \in \mathbb{R}^p$, where $f : \mathbb{R}^d \to \mathbb{R}, y_i \in \{\pm 1\}$. We call the function $l : \mathbb{R} \to \mathbb{R}, l(u) := e^{-\Phi(u)}$, exponentially tailed, if the following conditions hold:*

- *(i) $\Phi$ is continuously differentiable.*

- *(ii) $\Phi'(u) > 0$ for all $u \in \mathbb{R}$.*

- *(iii) The function $g(u) = \Phi'(u)u$ is non-decreasing in $[0, \infty)$.*

Notice that the definition above covers the exponential loss for $\Phi(u) = u$ and the logistic loss for $\Phi(u) = -\log\log(1 + e^{-u})$. To accommodate different loss functions, Assumption A3 needs to be adjusted as follows:

- There is a $t_0 > 0$, such that $\mathcal{L}(\boldsymbol{\theta}_{t_0}) < e^{-\Phi(0)}$.

See Section A in (Lyu & Li, 2020) for a more general, albeit technical, definition that allows the extension of the full analysis to general exponentially-tailed losses.

Under these conditions, we can define the soft margin as follows:

$$\widetilde{\gamma} = \frac{l^{-1}(\mathcal{L})}{\|\boldsymbol{\theta}\|^L} = \frac{\Phi^{-1}\left(\log\frac{1}{\mathcal{L}}\right)}{\|\boldsymbol{\theta}\|^L},$$

and prove a strict generalization of Theorem 3.1.

**Theorem A.22** (Soft margin increases - general loss function). *For almost any $t > t_0$, it holds:*

$$\frac{d\log\widetilde{\gamma}}{dt} \geq L\left\|\frac{d\boldsymbol{\theta}}{dt}\right\|^2 \left(\frac{(\Phi^{-1})'\left(\log\frac{1}{\mathcal{L}(\boldsymbol{\theta}_t)}\right)}{L\mathcal{L}(\boldsymbol{\theta}_t)\Phi^{-1}\left(\log\frac{1}{\mathcal{L}(\boldsymbol{\theta}_t)}\right)} - \frac{1}{\|\boldsymbol{\theta}_t\|\left\|\frac{d\boldsymbol{\theta}}{dt}\right\|}\right) \geq 0.$$

*Proof.* Let $\mathbf{n}_t \in \partial\|\boldsymbol{\theta}_t\|$. We have:

$$
\begin{aligned}
\frac{d\log\widetilde{\gamma}}{dt} &= \frac{d}{dt}\Phi^{-1}\left(\log\frac{1}{\mathcal{L}(\boldsymbol{\theta}_t)}\right) - L\frac{d}{dt}\log\|\boldsymbol{\theta}_t\| \\
&= \frac{d}{dt}\Phi^{-1}\left(\log\frac{1}{\mathcal{L}(\boldsymbol{\theta}_t)}\right) - L\left\langle\frac{\mathbf{n}_t}{\|\boldsymbol{\theta}_t\|}, \frac{d\boldsymbol{\theta}}{dt}\right\rangle \quad \text{(Chain rule)} \\
&\geq \frac{d}{dt}\Phi^{-1}\left(\log\frac{1}{\mathcal{L}(\boldsymbol{\theta}_t)}\right) - L\frac{\left\|\frac{d\boldsymbol{\theta}}{dt}\right\|}{\|\boldsymbol{\theta}_t\|} \quad \text{(definition of dual norm and } \|\mathbf{n}_t\|_\star \leq 1) \\
&= -\frac{d\mathcal{L}(\boldsymbol{\theta}_t)}{dt}\frac{(\Phi^{-1})'\left(\log\frac{1}{\mathcal{L}(\boldsymbol{\theta}_t)}\right)}{\mathcal{L}(\boldsymbol{\theta}_t)\Phi^{-1}\left(\log\frac{1}{\mathcal{L}(\boldsymbol{\theta}_t)}\right)} - L\frac{\left\|\frac{d\boldsymbol{\theta}}{dt}\right\|}{\|\boldsymbol{\theta}_t\|} \quad \text{(Chain rule)} \\
&= \left\|\frac{d\boldsymbol{\theta}}{dt}\right\|^2 \left(\frac{(\Phi^{-1})'\left(\log\frac{1}{\mathcal{L}(\boldsymbol{\theta}_t)}\right)}{\mathcal{L}(\boldsymbol{\theta}_t)\Phi^{-1}\left(\log\frac{1}{\mathcal{L}(\boldsymbol{\theta}_t)}\right)} - \frac{L}{\|\boldsymbol{\theta}_t\|\left\|\frac{d\boldsymbol{\theta}}{dt}\right\|}\right) \quad \text{(eq. Equation 18)}.
\end{aligned}
\tag{68}
$$

But, the first term inside the parenthesis can be related to the second one via the following calculation. Recall that, by the chain rule for locally Lipschitz functions (Theorem A.2), for any $\mathbf{g}_t \in \partial\mathcal{L}(\boldsymbol{\theta}_t)$ there exist $\mathbf{h}_1 \in \partial y_1 f(\mathbf{x}_1;\boldsymbol{\theta}_t), \ldots, \mathbf{h}_m \in \partial y_m f(\mathbf{x}_m;\boldsymbol{\theta}_t)$ such that $\mathbf{g}_t = \sum_{i=1}^{m} e^{-\Phi(y_i f(\mathbf{x}_i;\boldsymbol{\theta}_t))}\Phi'(y_i f(\mathbf{x}_i;\boldsymbol{\theta}_t)))\mathbf{h}_i$. Thus, for a minimum norm subderivative $\mathbf{g}_t^\star$, we have:

$$
\begin{aligned}
\langle\boldsymbol{\theta}_t, -\mathbf{g}_t^\star\rangle &= \left\langle\boldsymbol{\theta}_t, \sum_{i=1}^{m} e^{-\Phi(y_i f(\mathbf{x}_i;\boldsymbol{\theta}_t))}\Phi'(y_i f(\mathbf{x}_i;\boldsymbol{\theta}_t))\mathbf{h}_i^\star\right\rangle \\
&= \sum_{i=1}^{m} e^{-\Phi(y_i f(\mathbf{x}_i;\boldsymbol{\theta}_t))}\Phi'(y_i f(\mathbf{x}_i;\boldsymbol{\theta}_t))\langle\boldsymbol{\theta}_t, \mathbf{h}_i^\star\rangle \\
&= L\sum_{i=1}^{m} e^{-\Phi(y_i f(\mathbf{x}_i;\boldsymbol{\theta}_t))}\Phi'(y_i f(\mathbf{x}_i;\boldsymbol{\theta}_t))y_i f(\mathbf{x}_i;\boldsymbol{\theta}_t),
\end{aligned}
\tag{69}
$$

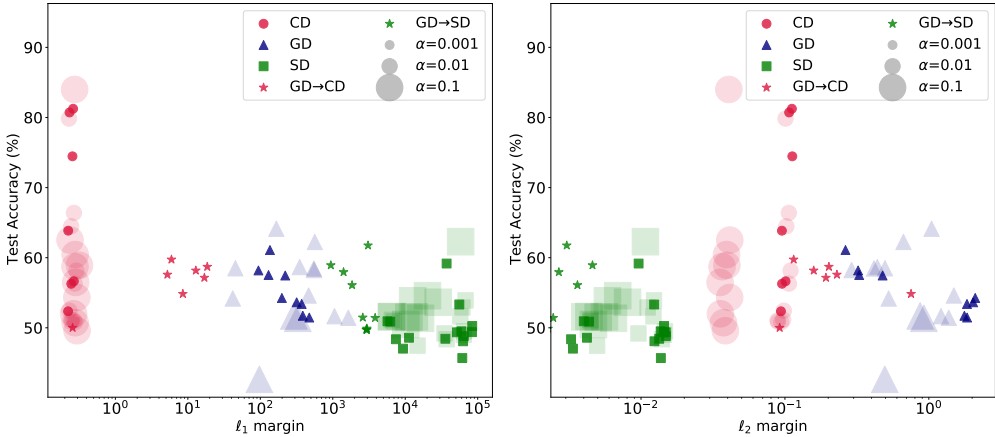

Figure 4: **Geometric margins vs test accuracy in a teacher-student setup.** *Left:* $\ell_1$ margin. *Right:* $\ell_2$ margin. Each point corresponds to a different run (different random seed).

where the last equality follows from Euler's theorem for homogeneous functions (whose generalization for subderivatives can be found in Theorem B.2 in Lyu & Li (2020)). But, now observe that as per assumption, $u \to \Phi'(u)u$ is non-decreasing and this last term can be lower bounded as:

$$\langle \boldsymbol{\theta}_t, -\mathbf{g}_t^\star \rangle \geq L \sum_{i=1}^{m} e^{-\Phi(y_i f(\mathbf{x}_i; \boldsymbol{\theta}_t))} \Phi' \left( \Phi^{-1} \left( \log \frac{1}{\mathcal{L}(\boldsymbol{\theta}_t)} \right) \right) \Phi^{-1} \left( \log \frac{1}{\mathcal{L}(\boldsymbol{\theta}_t)} \right), \tag{70}$$

where we used the fact $y_i f(\mathbf{x}_i; \boldsymbol{\theta}_t) \leq \Phi^{-1} \left( \log \frac{1}{\mathcal{L}(\boldsymbol{\theta}_t)} \right)$ for all $i \in [m]$ (by the monotonicity of $\Phi$ the definition of $\mathcal{L}$). Leveraging the fundamental property between the derivative of a function and its inverse's, we further get:

$$\langle \boldsymbol{\theta}_t, -\mathbf{g}_t^\star \rangle \geq L \sum_{i=1}^{m} e^{-\Phi(y_i f(\mathbf{x}_i; \boldsymbol{\theta}_t))} \frac{\Phi^{-1} \left( \log \frac{1}{\mathcal{L}(\boldsymbol{\theta}_t)} \right)}{(\Phi^{-1})' \left( \log \frac{1}{\mathcal{L}(\boldsymbol{\theta}_t)} \right)} = L\mathcal{L}(\boldsymbol{\theta}_t) \frac{\Phi^{-1} \left( \log \frac{1}{\mathcal{L}(\boldsymbol{\theta}_t)} \right)}{(\Phi^{-1})' \left( \log \frac{1}{\mathcal{L}(\boldsymbol{\theta}_t)} \right)}. \tag{71}$$

We have made the first term of eq. Equation 68 appear. By plugging eq. Equation 71 into eq. Equation 68, we get:

$$\begin{aligned} \frac{d \log \widetilde{\gamma}}{dt} &\geq \left\| \frac{d\boldsymbol{\theta}}{dt} \right\|^2 \left( \frac{L}{\langle \boldsymbol{\theta}_t, -\mathbf{g}_t^\star \rangle} - \frac{L}{\|\boldsymbol{\theta}_t\| \left\| \frac{d\boldsymbol{\theta}}{dt} \right\|} \right) \\ &\geq \left\| \frac{d\boldsymbol{\theta}}{dt} \right\|^2 \left( \frac{L}{\|\boldsymbol{\theta}_t\| \|\mathbf{g}_t^\star\|_\star} - \frac{L}{\|\boldsymbol{\theta}_t\| \left\| \frac{d\boldsymbol{\theta}}{dt} \right\|} \right) \quad \text{(definition of dual norm)}. \end{aligned} \tag{72}$$

Noticing that $\|\mathbf{g}_t^\star\|_\star = \left\| \frac{d\boldsymbol{\theta}}{dt} \right\|$ (from eq. Equation 19) concludes the proof. $\qquad\square$

## B  BREGMAN PROXIMITY MEASURE

In the proof of our main result (Theorem A.18), we constructed a sequence of approximate generalized KKT points (Definition A.9). However, in many cases, while solving (non-convex) optimization problems, we only have feasible points without any evidence of optimality or stationarity. In such cases, it is useful to come up with a *progress measure* of approximate stationarity that can also serve as a stopping criterion for the optimization algorithm. Consider an optimization problem:

$$\begin{aligned} \min_{\boldsymbol{\theta} \in \mathbb{R}^d} \ & f(\boldsymbol{\theta}) \\ \text{s.t. } & g_i(\boldsymbol{\theta}) \leq 0 \ \forall i \in [m], \end{aligned} \tag{P}$$

where we assume that $f, \{g_i\}_{i=1}^m$ are differentiable for the sake of brevity. For a feasible point $\boldsymbol{\theta}'$ of (P) and a non-negative function $d : \mathbb{R}^p \times \mathbb{R}^p \to \mathbb{R}_+$, we define the $d$-*Bregman proximity measure* as the solution of the following optimization problem:

$$\min_{\epsilon, \lambda_1, \ldots, \lambda_m} \epsilon$$

$$\text{s.t. } d\left(-\sum_{i=1}^m \lambda_i \nabla g_i(\boldsymbol{\theta}'), \nabla f(\boldsymbol{\theta}')\right) \leq \epsilon,$$

$$\sum_{i=1}^m \lambda_i g_i(\boldsymbol{\theta}') \geq -\epsilon, \tag{73}$$

$$\lambda_i \geq 0 \ \forall i \in [m].$$

This definition mirrors and generalizes the definitions of (Dutta et al., 2013), which were inspired by approximate KKT points (whose proximity is measured using the Euclidean distance as $d$). However, as we saw in our analysis, there are many cases of problems where a proximity measure would be better defined using alternatives functions. Figure 3 conceptually illustrates the reduction of the Bregman divergence in a possible converging path. The relaxation of the slackness constraints in the form of $\sum_{i=1}^m \lambda_i g_i(\boldsymbol{\theta}') \geq -\epsilon$ is essential for ensuring that the proximity measure captures proximity to stationarity - see Section 3.2 in (Dutta et al., 2013) for a discussion.

## C    RELATIONSHIP TO ADAM AND SHAMPOO

The family of steepest descent algorithms includes simplified versions (momentum turned-off) of two adaptive methods, Adam and Shampoo, which have been very popular for training deep neural networks.

### C.1    ADAM

Adam (Kingma & Ba, 2015) is a popular adaptive optimization method, which is frequently used in deep learning. Following our previous notation, the update rule of Adam amounts to:

$$\mathbf{m}_t = \beta_1 \mathbf{m}_{t-1} + (1 - \beta_1) \nabla \mathcal{L}(\boldsymbol{\theta}_{t-1})$$

$$\mathbf{v}_t = \beta_2 \mathbf{v}_{t-1} + (1 - \beta_2) \nabla \mathcal{L}(\boldsymbol{\theta}_{t-1})^2$$

$$\hat{\mathbf{m}}_t = \frac{\mathbf{m}_t}{1 - \beta_1^t}; \hat{\mathbf{v}}_t = \frac{\mathbf{v}_t}{1 - \beta_2^t} \tag{74}$$

$$\boldsymbol{\theta}_t = \boldsymbol{\theta}_{t-1} - \eta_t \frac{\hat{\mathbf{m}}_t}{\sqrt{\hat{\mathbf{v}}_t} + \epsilon},$$

where the $\sqrt{\cdot}, ^2, \div$ operations are overloaded to operate elementwise in vectors. Parameters $\beta_1, \beta_2$ control the memory of the update rule, while $\epsilon$ is a numerical precision parameter. Notice that for $\beta_1 = \beta_2 = \epsilon = 0$, we recover sign-gradient descent.

Wang et al. (2022)studied the implicit bias of (74) for $\epsilon > 0$ in linear networks establishing bias towards $\ell_2$ margin maximization, while Zhang et al. (2024) analyzed the case of $\epsilon = 0$ and generic $\beta_1, \beta_2 \in [0, 1)$ also in linear networks and showed bias towards $\ell_1$ margin maximization.

### C.2    SHAMPOO

Shampoo (Gupta et al., 2018) is an adaptive optimization algorithm, which has recently gained popularity in deep learning applications. For each weight matrix $\mathbf{W}_t$ and its corresponding gradient matrix $\mathbf{G}_t$, the update rule of Shampoo without momentum amounts to (Bernstein & Newhouse, 2024):

$$\mathbf{W}_{t+1} = \mathbf{W}_t - \eta_t \mathbf{U}_t \mathbf{V}_t^T, \tag{75}$$

where $\mathbf{U}_t, \mathbf{V}_t$ contain the left and right singular vectors of $\mathbf{G}_t$, i.e., $\mathbf{G}_t = \mathbf{U}_t \boldsymbol{\Sigma}_t \mathbf{V}_t$. Bernstein & Newhouse (2024) recently noticed that this update corresponds to steepest descent (in matrix space) with respect to the *spectral norm* $\sigma_{\max}(\cdot)$. This is equivalent to an architecture-dependent norm in parameter space. For instance, if $\boldsymbol{\theta} = (\mathbf{W}_1, \ldots, \mathbf{W}_L)$, then Shampoo without momentum corresponds to steepest descent with respect to the norm $\|\boldsymbol{\theta}\|_S := \max(\sigma_{\max}(\mathbf{W}_1), \ldots, \sigma_{\max}(\mathbf{W}_L))$.

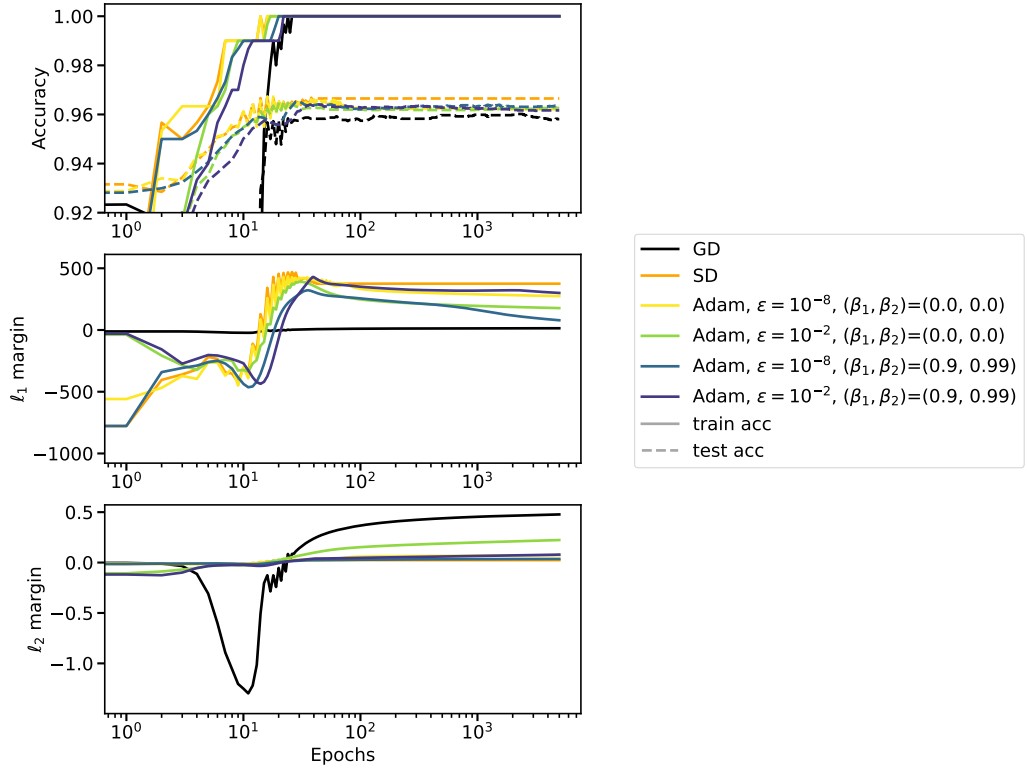

Figure 5: **Relationship between Adam and steepest descent algorithms.** Digits '2' and '7'.

## D EXPERIMENTAL DETAILS

All experiments are implemented in PyTorch (Paszke et al., 2017).

**Teacher-student experiments** We use the following hyperparameters: $d = 2^{32}, k = 64, k' = 1024, m = 250$, learning rate $\eta = 6 \times 10^{-3}$ and density $\frac{\|\theta^\star\|_0}{k'(d+1)} = 0.0001$ (3 coordinates active per neuron). We vary the scale of initialization in $\{0.1, 0.01, 0.001\}$ and we train for $10^5$ epochs. Each random seed affects the draw of the datasets and the initialization of the parameters of the network. Test accuracy is estimated using 20,000 unseen data drawn from the same generative process.

**MNIST** We use a constant learning rate of $3 \times 10^{-3}$ and 1-hidden layer neural networks of width 128, optimizing the logistic loss. The digits that we extract are '3' and '6' (100 training points). Each random seed corresponds to a different draw of the training dataset and different initialization. Sign gradient descent runs were very effective in minimizing the training loss, and we stopped the training early after the loss reached value smaller than $10^{-7}$ in order to avoid numerical issues. We depict the final value, repeated for as many epochs as shown in the figures (as if the model has indeed converged).

Figure 5 shows accuracy and margins for a different pair of digits ('2' vs '7').

