# OpenReview forum: "Flavors of Margin: Implicit Bias of Steepest Descent in Homogeneous Neural Networks"
_ICLR.cc/2025/Conference — ICLR 2025 Poster_

### Official Review · Reviewer_pCcn · 2024-11-01

**Soundness:** 3
**Presentation:** 3
**Contribution:** 2
**Rating:** 6
**Confidence:** 4

**Summary:**

This paper studies the implicit bias of the steepest flow with respect to a general norm in deep homogeneous neural networks. Theoretically, they present two major results. 1) The soft margin will increase 2) The limit points of the parameter directions are general KKT points of the margin maximization optimization problem under some assumption of the norm. They also show some experimental results concerning the trajectories of different steepest descent algorithms and the connection to the implicit bias of Adam.

**Strengths:**

This paper extends the result in [Lyu & Li 2020] to steepest flow for general norms with the lightest extra assumption on the norm. While previous papers focused on gradient descent and other variants of gradient descent, this paper gives a general result for any steepest flow.
The claimed results seem sound and the paper is nicely written.

**Weaknesses:**

The paper asserts that it presents results for the steepest descent; however, the theoretical results in Section 3 are limited to steepest flow. According to Lyu & Li (2020), additional work is required to extend the findings from flow to descent.

Regarding the general loss function, the paper demonstrates that the soft margin is monotonically increasing but fails to address the convergence of KKT points.

Moreover, the paper overlooks several relevant references, including:

1. Chatterji, N.S., Long, P.M., & Bartlett, P. (2021). "When does gradient descent with logistic loss interpolate using deep networks with smoothed ReLU activations?"

2. Cai, Y., Wu, J., Mei, S., Lindsey, M., & Bartlett, P.L. (2024). "Large Stepsize Gradient Descent for Non-Homogeneous Two-Layer Networks: Margin Improvement and Fast Optimization."

3. Nacson, M.S., Srebro, N., & Soudry, D. (2019, April). "Stochastic gradient descent on separable data: Exact convergence with a fixed learning rate."

4. Ji, Z., Srebro, N., & Telgarsky, M. (2021, July). "Fast margin maximization via dual acceleration."

**Questions:**

- Can the authors extend the existing results on steepest flow to steepest descent? If not, it would be more accurate to clarify that the current theoretical results apply solely to steepest flow.

- Is it feasible to demonstrate KKT point convergence for the general loss function?

- In Lyu & Li (2020), tight convergence rates for both the loss function and parameter norms were established. Are there any mathematical challenges in achieving similar results here? Additionally, they showed that the soft margin approximates the normalized margin. Could you provide a similar analysis in this context?

---

> ### Author Response · Authors · 2024-11-20
> **Authors' response (1/2)**
>
> Thank you very much for the critical review of our work and your help in improving it!
>
> First, before responding to your questions, please allow us to correct a small mistake in your evaluation of our work, in order to avoid confusions:
>
> > The limit points of the parameter directions are general KKT points of the margin maximization optimization problem under some assumption of the norm
>
> Assumming that by "some assumption of the norm" you mean the algorithm norm squared being smooth, then: Theorem 3.8, which is the main result of this paper, asserts that steepest flow converges in direction to a generalized KKT point *without* any assumption on the algorithm norm. Corollary 3.8.1, on the other hand, strengthens this result in the case of a norm that satisfies the aforementioned property, stating that we get convergence to a *(standard) KKT point*, not a generalized one.
>
> > Can the authors extend the existing results on steepest flow to steepest descent? If not, it would be more accurate to clarify that the current theoretical results apply solely to steepest flow.
>
> Indeed, our paper focuses on steepest descent with infinitesimal learning rate, i.e., steepest flow (Eq. 3). We specify this at the start of our introduction (line 051). We leave the generalization of our results to the case of steepest descent (with a non infinitesimal learning rate) as an open challenge. Notice that the possibility of a non-smooth norm substantially complicates a discrete time analysis - for instance, Theorem E.2 and Lemmata E.7, E.8, E.9 in [Lyu & Li, 2020] only hold in the case of an $\ell_2$ geometry. Furthermore, the proof will have to operate under more restrictive assumptions on the smoothness of the network, which, arguably, minimizes the practical relevance of such a result. We recognize this as a limitation of our analysis and added a short note on this in our conclusion (Section 5, line 478).
>
> > Is it feasible to demonstrate KKT point convergence for the general loss function?
>
> We believe that this should be possible to prove for any exponentially-tailed loss. In particular, in Appendix A.3, we demonstrate how Theorem 3.1 generalizes to other loss functions, and we note that a more technical definition than Definition A.21 could allow the extension of our full proof to other loss functions, similar to the approach taken in [Lyu & Li, 2020]. As a matter of fact, in the Experiments section (Section 4), we consider training with both the exponential and logistic loss with various steepest descent algorithms, and the margin observations are consistent with our theory. Finally, let us remark that it is a common practice in the literature to focus solely on the exponential loss (see, for instance, [1, 2, 3]) as it is known to capture the essence of these results.
>
> 1. Implicit Bias in Deep Linear Classification: Initialization Scale vs Training Accuracy. Edward Moroshko, Suriya Gunasekar, Blake Woodworth, Jason D. Lee, Nathan Srebro, Daniel Soudry.
> 2. Implicit Bias of Gradient Descent on Linear Convolutional Networks. Suriya Gunasekar, Jason Lee, Daniel Soudry, Nathan Srebro.
> 3. Lexicographic and Depth-Sensitive Margins in Homogeneous and Non-Homogeneous Deep Models. Mor Shpigel Nacson, Suriya Gunasekar, Jason D. Lee, Nathan Srebro, Daniel Soudry.
>
> > In Lyu & Li (2020), tight convergence rates for both the loss function and parameter norms were established. Are there any mathematical challenges in achieving similar results here?
>
> No, we believe there are no mathemetical challenges in achieving similar complementary results. Essentially, Lemma A.6 in our paper characterizes the rate of convergence for the loss and the paramater norm (particularly for the relevant algorithm norm). We opted not to risk presenting such a complimentary result during the rebuttal period (after the first round of all the reviews), since it is not exactly straightforward, but we believe that an interested reader can deduce these rates using our Lemma A.6 and equations 30 and 31.
>
> > Additionally, they showed that the soft margin approximates the normalized margin. Could you provide a similar analysis in this context?
>
> If we understand your question correctly, then, yes, this is (essentially) proved in Lemmata A.6 and A.7. We fixed a minor typo there (line 822) and added the conclusion as a corollary (Corollary A.7.1). Furthermore, we added a relevant pointer for this in the main text (lines 178-179). Thank you very much for the suggestion!

---

> > ### Author Response · Authors · 2024-11-20
> > **Authors' response (2/2)**
> >
> > > Moreover, the paper overlooks several relevant references, including:
> > Chatterji, N.S., Long, P.M., & Bartlett, P. (2021). "When does gradient descent with logistic loss interpolate using deep networks with smoothed ReLU activations?"
> > Cai, Y., Wu, J., Mei, S., Lindsey, M., & Bartlett, P.L. (2024). "Large Stepsize Gradient Descent for Non-Homogeneous Two-Layer Networks: Margin Improvement and Fast Optimization."
> > Nacson, M.S., Srebro, N., & Soudry, D. (2019, April). "Stochastic gradient descent on separable data: Exact convergence with a fixed learning rate."
> > Ji, Z., Srebro, N., & Telgarsky, M. (2021, July). "Fast margin maximization via dual acceleration."
> >
> > Thank you for the references. We added a discussion of non-homogeneous networks in lines 094-096 and the revised version now cites paper #2. However, we believe that papers #1, #3 and #4 are sligthly peripheral to our paper. Please note that we already included a reference to a comprehensive survey on implicit bias [Vardi, 2023], where some of these papers are discussed in depth. If you disagree or have suggestions on where it would be helpful to discuss these works in our paper, we would be happy to hear them.
> >
> > We would like to thank you once again for your constructive critism. Please let us know if any of our answers is unclear or if you have further questions. We hope that our explanations, together with the manuscript changes made in response to your and the other reviewers' comments, could perhaps encourage you to raise your evaluation score of our work. Thank you!

---

> > > ### Comment · Reviewer_pCcn · 2024-11-27
> > >
> > > Thank you for your response. I would like to keep my score.

---

> > > > ### Author Response · Authors · 2024-11-28
> > > >
> > > > Thank you!

---

### Official Review · Reviewer_2hLy · 2024-11-03

**Soundness:** 2
**Presentation:** 2
**Contribution:** 2
**Rating:** 6
**Confidence:** 2

**Summary:**

This work studies how the prediction margin changes as the training proceeds in a homogeneous network

**Strengths:**

The problem of prediction margin is important and of fundamental importance

**Weaknesses:**

I am not sure if I understand the meaning of the result. Also, I find the setting of the theory too restrictive

1. I do not understand Figure 1-right. What does it imply? How does this relate to the theory?
2. The theory only applies to the "exponential loss," which does not make much sense to me.

Lastly, I feel I should comment that I do not have the related background knowledge to assess the technical advancement made by this work

**Questions:**

How close is the steepest descent to actual GD? Does the theory hold for GD?

---

> ### Author Response · Authors · 2024-11-20
> **Authors' response**
>
> Thank you for reading our paper and helping us improve it!
>
> We reply to your questions:
>
> > I do not understand Figure 1-right. What does it imply? How does this relate to the theory?
>
> Thank you for the question! In Figure 1 (right), we compare 3 different steepest descent algorithms: gradient descent, sign gradient descent and coordinate descent. We measure the $\ell_\infty$ margin of the training dataset at the end of training (Eq. 14) and plot it against the test accuracy of the networks (each point -- circle, square, triangle -- corresponds to a different seed). The first observation, which connects to Theorem 3.1, is that coordinate descent has a larger $\ell_\infty$ margin than the rest of the algorithms -- see lines 409-412. This is expected since it is the only algorithm out of the 3 which is guaranteed to increase this margin during training. The second observation is about the connection between margin and generalization for this task and is less related to our theory. In short, we comment on the fact that these seems to be no strong, causal link between larger margin and generalization in this setting - see lines 412-425.
>
> > The theory only applies to the "exponential loss," which does not make much sense to me.
>
> This is a valid concern, but our choice of this loss follows a long list of precedence and justification in the literature. Indeed, the reason why many papers are concerned with the exponential or exponentially-tailed losses is that: (a) they are close to the most common choice of the cross-entropy loss, or its binary version, the logistic loss, and (b) they allow for a clean relationship between the geometric margin and the loss via the softmax. Also, let us mention that this has been a standard practice in the literature, with numerous papers (e.g. [1, 2, 3]) that **only** study training under the exponential loss. In defense of the contributions of our paper, let us note that our work considers the generalization to other losses in Section A.3 (where Theorem 3.1 is generalized).
>
> > How close is the steepest descent to actual GD? Does the theory hold for GD?
>
> A steepest descent algorithm is defined with respect to a norm. When this norm is the $\ell_2$, we obtain gradient descent. Otherwise, we obtain a different algorithm. The theory holds for gradient descent with infinitesimal learnging rate, but this is not the most interesting case since the result of [Lyu & Li, 2019] already covers this case.
>
> We hope we were able to answer your questions convicingly. Please let us know if there are any other concerns or if you require further explanations.
>
> In general, given your comment:
> > Lastly, I feel I should comment that I do not have the related background knowledge to assess the technical advancement made by this work
>
> We would be happy to help you assess the technical advancements of this paper by explaining things further if you feel this is appropriate, in order to provide a basis that would allow you to raise your score and recommend acceptance. Thank you for your time and your help!
>
>
> 1. Implicit Bias in Deep Linear Classification: Initialization Scale vs Training Accuracy. Edward Moroshko, Suriya Gunasekar, Blake Woodworth, Jason D. Lee, Nathan Srebro, Daniel Soudry.
> 2. Implicit Bias of Gradient Descent on Linear Convolutional Networks. Suriya Gunasekar, Jason Lee, Daniel Soudry, Nathan Srebro.
> 3. Lexicographic and Depth-Sensitive Margins in Homogeneous and Non-Homogeneous Deep Models. Mor Shpigel Nacson, Suriya Gunasekar, Jason D. Lee, Nathan Srebro, Daniel Soudry.

---

> > ### Comment · Reviewer_2hLy · 2024-11-25
> > **reply and thoughts**
> >
> > Thanks for the rebuttal. I feel more positive towards the paper and have raise my rating to 6. I thought steepest descent is more restrictive than GD, but now it looks like it contains GD as a subclass, which is nice
> >
> > However, I point out that my rating is at best an educated guess.

---

> > > ### Author Response · Authors · 2024-11-26
> > >
> > > Thank you for your time and engagement during the rebuttal period!

---

### Official Review · Reviewer_By2M · 2024-11-05

**Soundness:** 3
**Presentation:** 3
**Contribution:** 3
**Rating:** 6
**Confidence:** 3

**Summary:**

This paper studies the implicit bias of  steepest flow with general norm in homogeneous neural networks. It shows that the soft-margin is monotonically increasing after the loss is below a threshold. Furthermore, it shows that the steepest flow converges to the generalized KKT point. Finally, experiments demonstrate the similarity of Adam algorithm and Signed-steepest descent algorithm, in terms of implicit bias.

**Strengths:**

1. The convergence of the general steepest descent to the general maximum margin solution is novel.
2. The connection between the implicit bias of Signed steepest descent with Adam is interesting.

**Weaknesses:**

The set of results and proofs seem to be a straightforward generalization of [Lyu and Li, 2019]. The technical contribution is not strong.

**Questions:**

See weakness

---

> ### Author Response · Authors · 2024-11-20
> **Authors' response**
>
> We appreciate the time you took to review our paper and are grateful you found our results novel! We reply to your only question regarding the technical contributions of the paper:
>
> > The set of results and proofs seem to be a straightforward generalization of [Lyu and Li, 2019]. The technical contribution is not strong.
>
> While this is of course a relative matter, please let us elaborate on two technical points where our work departs significantly from [Lyu & Li, 2019]:
> - Non-smoothness of the algorithm norm: the fact that the squared norm of the algorithm is not necessarily continuously differentiable (as it is in the case of gradient descent) complicates the analysis in many parts - see, for example, Proposition A.3 and the rest of the analysis where specific care must be taken to handle updates aligned with the least norm subgradient $\mathbf{g}_t^\star$.
> - Quantifying proximity to stationarity: Gradient descent approaches the KKT points of the $\ell_2$ max margin problem by reducing a Euclidean distance measuring stationarity. Prior to our work, it was arguably unclear how a different steepest descent method might have progressed to approach stationarity of the corresponding implicit max margin problem. We introduced a novel measure of stationarity, based on an algorithm-specific (pseudo) Bregman divergence, and we show that this succeeds in quantifying the progress of the algorithms. This also naturally motivates a new progress measure for approximate stationarity (Section B), which could be of independent interest. Notice that here as well the non-smoothness of the norm causes technical challenges, and their handling required some fairly involved machinery from Convex Analysis (Proposition A.13 & Theorem 5.24 in [Beck, 2017]).
>
> In general, the proof of [Lyu & Li, 2019] is heavily based on the $\ell_2$ geometry of gradient descent (naturally so!) and many parts have to be modified in order to obtain a clean result for any steepest flow. Testament to this are the many novel tools (generalized KKT points -- Definition 3.4, approximate generalized KKT points -- Definition A.9, generalized Bregman divergences -- Definition 3.6) and corresponding results (e.g. Proposition A.20) which we had to introduce in order to obtain our implicit bias characterization.
>
> We would be happy if our explanations could make you reconsider the technical contribution of this work, and perhaps, make you raise your score. Please let us know if you have any specific questions. Thank you!

---

> > ### Author Response · Authors · 2024-11-29
> > **Kind reminder**
> >
> > Hi,
> >
> > We would like to kindly encourage you to respond to our comments, as the discussion period ends soon.
> >
> > Thank you!

---

### Official Review · Reviewer_APGr · 2024-11-05

**Soundness:** 4
**Presentation:** 4
**Contribution:** 3
**Rating:** 8
**Confidence:** 5

**Summary:**

This paper studies the implicit bias of steepest descent methods with infinitesimal step size (i.e., steepest flow) and presents an interesting generalization of Lyu & Li (2020)'s result on gradient flow (steepest flow with L2 norm). Similar to gradient flow, it is shown that steepest flow has a bias towards KKT solutions to a margin maximization problem, but the margin being maximized here is normalized properly according to the norm used in steepest flow, which may not be the L2 norm in general.

**Strengths:**

1. Understanding the implicit bias of optimization methods is an important problem in deep learning theory, and the paper takes a margin-based view of implicit bias and sheds light on how different methods may optimize different notions of margin for deep models.
2. It is a bit surprising that such an implicit bias of steepest descent can be rigorously proved in the general case. Previously, it is known that this can be proved for linear models, and for deep homogeneous networks, Lyu & Li (2020) generalized the result for GD. It is reasonable to imagine that one could somehow generalize Lyu & Li (2020)'s result to steepest gradient, but their proof apparently has a lot of dependence on the L2 geometry induced by GD. After having a careful read of the proof (though I skipped a few details), I believe the authors found the right way to do this generalization and worked out all the technical issues, including those related to non-smoothness.
3. The paper is well-written and easy to follow.
4. Experiments in simple settings validated the theoretical results.

**Weaknesses:**

1. While I really appreciate that the authors worked out every technical detail to generalize the proof of Lyu & Li (2020), I also noted that the overall proof outline has not changed much from Lyu & Li (2020), which means this paper does not actually bring a brand new high-level proof idea. This is reasonable since any implicit bias analysis of steepest flow automatically implies an analysis of gradient flow, but the fact that the paper fails to prove the real KKT conditions and only manage to prove a weaker notion (generalized KKT) suggests that a perfect generalization from gradient flow to steepest flow may indeed need a completely new proof strategy.
2. Same as all previous works studying the max-margin bias, this paper only analyzes the late phase, where the training loss is already very small. While the asymptotic analysis of implicit bias is interesting, this also makes the phenomenon less relevant to the practice.

Minor issue: there are large blanks on some pages in the appendix.

**Questions:**

This paper only proves the convergence to generalized KKT solutions for sign gradient flow, which is an important case of steepest descent. I wonder if the authors have any thoughts on whether failing to prove KKT in this case is an artifact due to the proof techniques or if some tricky hard cases actually exist.

---

> ### Author Response · Authors · 2024-11-20
> **Authors' response**
>
> We are really grateful for your careful and critical review and we are encouraged you found our paper well-written, easy to follow, and its contributions "surprising". In particular, we are glad that your thorough reading recognized that the [Lyu & Li, 2019] proof relies heavily on the $\ell_2$ geometry (which, of course, is natural to some extent), and our analysis is not straightforward.
>
> We respond to your questions:
>
> > Minor issue: there are large blanks on some pages in the appendix.
>
> Thank you. This is fixed now.
>
> > This paper only proves the convergence to generalized KKT solutions for sign gradient flow, which is an important case of steepest descent. I wonder if the authors have any thoughts on whether failing to prove KKT in this case is an artifact due to the proof techniques or if some tricky hard cases actually exist.
>
> We have indeed spent quite some time pondering this question. In short, it is unclear whether convergence to KKT points can be proven in general. We share some thoughts on this topic:
>
> For sign gradient descent ($\|\cdot\| = \|\cdot\|_\infty$), the Bregman (pseudo) divergence is defined with respect to the $\ell_1$ norm squared. Since the $\ell_1$ norm squared is not strictly convex, the divergence between two vectors approaching 0 (as shown in Proposition A.15 and Theorem A.18), does not imply that the two vectors converge to the same point. A fairly standard technique in convex analysis might suggest to consider a different divergence: the Bregman divergence induced by the (normalized) negative entropy, which is strongly convex with respect to the $\ell_1$ norm. As a result, the obtained divergence, which is a normalized version of the KL divergence, becomes a proper divergence. Unfortunately, two issues arise with this approach: (a) the KL divergence requires non-negative values, which remains problematic even after employing the common trick of doubling the dimension, and (b) if we have a sequence of KL-approximate KKT points, it is not clear whether this implies convergence to a KKT point (that is, it is unclear whether Proposition A.20 holds for d=KL). Had this approach been successful, we could apply a dual rationale for coordinate descent, using the divergence induced by the log-sum-exp function.
>
> One potential experimental approach to determine  whether our result is an artifact of our proof technique or not would involve leveraging Corollary A.19.1 and, in particular, eq. 60. There, we show that the rate of convergence to stationarity (for smooth squared norms) depends on the smoothness parameter of the norm. Thus, an experiment could involve training a homogeneous neural network using various steepest descent algorithms (e.g., $\ell_p$ norms with $p = {2, 3, 4}$) and observing whether this theoretical lower bound manifests. Specifically, it would be interesting to see if, once the training data are fit, convergence to stationarity takes longer for algorithms with less smooth norms. However, this may be challenging since measuring some of these quantities in practice could be difficult due to subgradients.
>
> In total, while we remain uncertain about the $\ell_1$ and $\ell_\infty$ cases, we believe that if convergence to a KKT point is true for any norm, a proof similar to ours would have been applicable. Although we do not currently have a definitive answer, we hope you find these insights interesting.
>
> Finally, we would kindly suggest taking a look at our global response and the revised version of the paper. In particular, we think you might find Section C.2 interesting, where we added some discussion on the relevance of our results to a newly introduced adaptive method. Thank you once again!

---

> > ### Comment · Reviewer_APGr · 2024-11-24
> >
> > Thanks so much for your thorough response. I would like to keep my high score.

---

> > > ### Author Response · Authors · 2024-11-24
> > >
> > > Thank you!

---

### Official Review · Reviewer_Bpd2 · 2024-11-11

**Soundness:** 4
**Presentation:** 3
**Contribution:** 3
**Rating:** 6
**Confidence:** 3

**Summary:**

The paper investigates the implicit bias of the family of steepest descent algorithms, including gradient descent, sign gradient descent, and coordinate descent, for deep homogeneous neural networks. The authors aim to understand how these optimization algorithms implicitly select solutions from the perspective of margin maximization. The main contributions of the paper are:

1. A rigorous analysis of the implicit bias of steepest descent algorithms in non-linear, homogeneous neural networks, showing that an algorithm-dependent geometric margin increases during training, under the same set of assumptions made in Lyu&Li.
2. The definition and characterization of a generalized notion of stationarity for optimization problems, demonstrating that these algorithms reduce a generalized Bregman divergence, which quantifies proximity to stationary points of a margin-maximization problem.
3. Experimental validation of the theoretical findings by training neural networks with various steepest descent algorithms, highlighting connections to the implicit bias of Adam.


I think the class of the optimization problems studied in this paper is general and the theoretical results of the paper are quite clean. Although the main proofs follow the framework developed by Lyu and Li, the proofs are nontrivial (I read most of the proofs in the main text and a few in the appendix). Overall, I am leaning towards accepting the paper. With additional clarifications of the relation with prior work, this paper would be a valuable addition to the literature on deep learning theory.

**Strengths:**

**Strong Points:**

1. The paper provides rigorous theoretical analysis of the implicit bias of steepest descent algorithms. The results and proofs extend the l2 norm case (gradient descent) by Lyu&Li to general norms. The theoretical results are clean and some of the extension are nontrivial.

2. The class of optimization algorithms studied in this paper is very general and include several important optimization algorithms as special cases.

3. The paper connects theoretical insights to practical optimization methods like Adam, which is widely used in the deep learning community. This connection may help to understand the benefit of Adam.

**Weaknesses:**

**Weak Points:**
The authors briefly discuss some prior work on implicit bias of optimization algorithms other than GD in the related work section. But I found the inclusion of related work and the discussion here is somewhat insufficient. For example, Gunasckar et al. 18 already studied the implicit bias of several class of optimization algorithms. However, the citation here is very superficial and it is unclear to me how the results in this paper supercede (or compared with) their results. It is also unclear to me how the results by Wang et al. on Adam is related to the results of this paper. If we simplify Adam to signGD, it seems that the results in Wang et al. is inconsistent with this paper? I think more detailed discussion is necessary.

Moreover, the following papers also obtain implicit bias results towards L_p margin with p different from 2 (for special models). Is there any concrete relation between these results and the results in this paper.
Kernel and Rich Regimes in Overparametrized Models (already cited but not discussed in details)
Implicit Bias in Deep Linear Classification: Initialization Scale vs Training Accuracy

**Questions:**

minor comment:
1. The sentence before Theorem 3.1. The proof is quite similar to that in Lyu&Li. It seems to me that this paper used Cauchy-shwartz but Lyu&Li explicitly used the Pythagorean theorem (which only holds in inner prod space but not general normed space). If I am correct, then there is not much difference.

2. I think it would be better to discuss briefly the relation between D-generalized KKT and ordinary KKT between Theorem 3.8 and Cor 3.8.1. and why not to prove something like Cor 3.8.1. directly.

---

> ### Author Response · Authors · 2024-11-20
> **Author's response (1/2)**
>
> Thank you very much for taking the time to review our paper and for helping us improve it! We reply to your comments:
>
> > Gunasckar et al. 18 already studied the implicit bias of several class of optimization algorithms. However, the citation here is very superficial and it is unclear to me how the results in this paper supercede (or compared with) their results.
>
> Gunasekar et al. (2018) studied the implicit bias of all steepest descent algorithms in *linear* models under the exponential loss. Our paper, in contrast, analyzes steepest descent in *homogeneous*, potentially non-linear, neural networks. So, our setting is more general than Gunasekar's et al. (2018). The strongest results we obtain in this paper characterize the limiting points of the trajectory as KKT points of the margin maximization problem (in the case of a smooth squared norm), whereas Gunasekar et al. (2018) prove that in linear models, we obtain global *maximizers* of the equivalent margin maximization problem. The gap between the two results is, however, unavoidable, as it is known that even for gradient descent, homogeneous networks can converge to points which are KKT points, but not necessarily (even) locally optimal with respect to the margin [Vardi et al. (2022)]. We view our results as conceptually consistent with those of Gunasekar et al. (2018), and we have added one sentence elaborating on this in line 087.
>
> > It is also unclear to me how the results by Wang et al. on Adam is related to the results of this paper. If we simplify Adam to signGD, it seems that the results in Wang et al. is inconsistent with this paper?
>
> Let us try clarify the confusion. Wang et al. (2021, 2022) analyzes the version of Adam where a precision paramater $\epsilon$ is added in the denominator to avoid numerical instability (see eq. 71 in our paper). In order to obtain signGD, we need to set $\epsilon$ to 0, so the results of Wang et al. (2021, 2022) **do not apply**. This issue was recently observed by Zhang et al. (2024), who analyze Adam without the precision parameter ($\epsilon=0$), but with momentum, specifically in **linear** models. As we mention in line 101, they "found bias towards l1 margin maximization - the same as in the case of sign gradient descent". Thus, our results are consistent with [Zhang et al. (2024)]. We elaborate on this topic further in Section 4.2, where we test these results empirically. As a reminder, in Section 4.2, we observe experimentally that models trained with Adam initially exhibit an $\ell_1$ bias, which later changes to an $\ell_2$ bias. We have updated the text in that section, corrected two minor typos that may have caused confusion, and revised Section C.1 in the Appendix, which discusses Adam in more detail.
>
> > The authors briefly discuss some prior work [...] I think more detailed discussion is necessary.
>
> Thank you for the suggestion. We have included additional discussion and references in that section.
>
> > Moreover, the following papers also obtain implicit bias results towards L_p margin with p different from 2 (for special models). Is there any concrete relation between these results and the results in this paper.
> > Kernel and Rich Regimes in Overparametrized Models (already cited but not discussed in details) Implicit Bias in Deep Linear Classification: Initialization Scale vs Training Accuracy
>
> Thank you for the question. These papers study gradient descent in a simple class of homogeneous networks (referred to as diagonal neural networks), defined as $f(\mathbf{x}; \mathbf{u}) $= $\langle$ $\mathbf{u}$_+^D - $\mathbf{u}$_-^D,$\mathbf{x} \rangle$, $D > 0$. The constant $D$ represents the depth of the network. The focus of these papers is on how the initialization scale and depth $D$ affect the implicit bias of optimization, along with a precise characterization of the transition from the kernel to the rich regime during training. In their setting, it is known that gradient descent has a bias towards minimum $\ell_2$ norm of the **parameters** [Lyu & Li, 2019], but these papers explore how this bias translates into the **predictor** (i.e. function) space (see Section 2 of [Woodworth et al, 2020. *Kernel and Rich Regimes in Overparametrized Models*]). In this predictor space, they prove an implicit bias towards norm minimization for a norm different than the Euclidean one. In contrast, our results focus solely on the parameter space of homogeneous networks trained with steepest descent. We view these works as peripheral to ours, which is why we opted not to discuss them in detail (as doing so would require defining implicit bias in both parameter and predictor spaces, which, in our view, would detract from the focus of the paper).

---

> ### Author Response · Authors · 2024-11-20
> **Author's Response (2/2)**
>
> > The sentence before Theorem 3.1. The proof is quite similar to that in Lyu&Li. It seems to me that this paper used Cauchy-shwartz but Lyu&Li explicitly used the Pythagorean theorem (which only holds in inner prod space but not general normed space). If I am correct, then there is not much difference.
>
> Yes, this is "conceptually" correct (Theorem 3.1 does not use the CS inequality, but rather the definition of the dual norm, while Lemma 5.1 in [Lyu & Li, 2019] does use a polar decomposition and the Pythagorean theorem). This is why we mentioned that Theorem 3.1 "is similar to part of Lemma 5.1 in [Lyu & Li, 2019]" in line 181.
>
> > I think it would be better to discuss briefly the relation between D-generalized KKT and ordinary KKT between Theorem 3.8 and Cor 3.8.1. and why not to prove something like Cor 3.8.1. directly.
>
> Thank you for this suggestion! We added a short discussion in lines 315-317.
>
> Please also consider taking a look at the general response to all reviewers. In particular, in addition to the improvements in presentation, the updated manuscript contains a small section on the connection between a newly introduced adaptive optimization algorithm (Shampoo) and our framework.
> We are grateful for your useful questions and suggestions, which helped us improve our paper. Please let us know if any of our answers is unclear or if you have further questions. If not, we kindly ask you to consider raising your score, which would help achieve a consensus among reviewers. Thank you!

---

> ### Comment · Reviewer_Bpd2 · 2024-11-25
>
> Thanks for the response. The response and the revision have improved the quality of the paper slightly. I am leaning towards accepting the paper and my current rating is somewhere around 6.5. While I might consider rounding it up to a 7 if that were an option, it may not reach the caliber of an 8 in my opinion (compared to other ICLR rating-8 papers I’ve reviewed and read, in terms of conceptual and technical novelty). Therefore, I will maintain my current rating.

---

> > ### Author Response · Authors · 2024-11-25
> >
> > Thank you for engaging during the rebuttal period! We are glad about your positive evaluation of our work.

---

### Author Response · Authors · 2024-11-20
**General Response**

We would like to thank all reviewers for their time and efforts in helping us improve our paper. We uploaded a revised version of the paper, which incorporates some of your suggestions (colored in red). In addition, we want to highlight two points:

1. We updated the Related work section to make the discussion of prior work more elaborate. Please consider taking a look.
2. We added some discussion on the connection between an adaptive optimization method, Shampoo [1], and steepest descent which was recently brought to our attention and discuss how it is related to the results of this paper (lines 130-132 and Appendix C.2).

Thank you!

[1] Vineet Gupta, Tomer Koren, and Yoram Singer. Shampoo: Preconditioned Stochastic Tensor Optimization.

---

### Meta-Review · Area_Chair_1acp · 2024-12-21

**Metareview:**

This paper studies the implicit bias of steepest flow (continuous-time version of steepest descent) algorithm in training homogeneous neural networks. The paper extends an existing work Lyu & Li on gradient descent to a general class of algorithms, showing monotonic increase of the soft margin and convergence to the direction of generalized KKT points of the margin maximization problem.

All reviewers were positive about this submission. The paper presents an extension of a seminal result to a much more general class of algorithms, also drawing connections to practical algorithms such as Adam and Shampoo. Although the outline of proof seems to be the same as the existing paper, the extension from $\ell_2$ geometry to possibly nonsmooth cases looks nontrivial.

It is, though, unfortunate that the current proof technique cannot guarantee convergence to the usual KKT points (not the generalized KKT points introduced in the paper) in some important cases such as $\ell_1$ and $\ell_\infty$ norms.

Also, as Reviewer pCcn points out, the paper only characterizes the implicit bias in the continuous-time (flow) case and leaves the extension to the discrete-time (descent) case as future work. In fact, I see that there is no indicator of this fact in the title and abstract; I recommend revising at least the abstract to reflect this.

Overall, the paper offers solid contributions to the study of implicit bias of neural network training. I recommend acceptance.

**Additional Comments On Reviewer Discussion:**

Other than the points mentioned above, noteworthy points about the discussion include:
- Some reviewers pointed out missing references and suggested improving the related works section; the authors made revisions accordingly.
- Reviewer pCcn asked if tight convergence rates for both the loss function and parameter norms can be established, as in Lyu and Li, and the authors responded that such rates can be deduced from their analysis. To me, it seems beneficial to state these convergence rate results in a more explicit form (in a proposition, etc).

---

### Decision · Program_Chairs · 2025-01-22

Accept (Poster)